

# Preliminary results from the FARCE 2015 campaign: multidisciplinary study of the forests-gases-aerosols-clouds system on the tropical island of La Réunion

Valentin Duflot[1,2], Pierre Tulet[1], Olivier Flores[3], Christelle Barthe[1], Aurélie Colomb[4], Laurent Deguillaume[4], Mickael Vaïtilingom[4,a], Anne Perring[5,6], Alex Huffman[7], Mark T. Hernandez[8], Karine Sellegri[4], Ellis Robinson[5,6], David J. O'Connor[7,b], Odessa M. Gomez[8], Frédéric Burnet[9], Thierry Bourrianne[9], Dominique Strasberg[3], Allan K. Bertram[10], Patrick Chazette[11], Julien Totems[11], Jacques Fournel[3], Pierre Stamenoff[2], Jean-Marc Metzger[2], Mathilde Chabasset[1], Clothilde Rousseau[1], Eric Bourrianne[1,c], Martine Sancelme[12], Anne-Marie Delort[12], Rachel E. Wegener[7], Cedric Chou[10], and Pablo Elizondo[10]

[1]Laboratoire de l'Atmosphère et des Cyclones (LACy), UMR8105, Université de la Réunion - CNRS - Météo-France, Saint-Denis de La Réunion, France
[2]Observatoire des Sciences de l'Univers de La Réunion (OSUR), UMS3365, Saint-Denis de la Réunion, France
[3]UMR PVBMT, Peuplements Végétaux et Bioagresseurs en Milieu Tropical, Saint-Denis de La Réunion, France
[4]Université Clermont Auvergne, CNRS Laboratoire de Météorologie Physique (LaMP), 63000 Clermont-Ferrand, France
[5]National Oceanic and Atmospheric Administration, Boulder, CO, USA
[6]Cooperative Institute for Research in Environmental Science, Boulder, CO, USA
[7]Department of Chemistry and Biochemistry, University of Denver, Denver, CO, USA
[8]Department of Civil and Environmental Engineering, UC Boulder, Boulder, CO, USA
[9]CNRM, Centre National de la Recherche Météorologique, UMR3589, CNRS - Météo-France, Toulouse, France
[10]Department of Chemistry, University of British Columbia, Vancouver, BC, Canada
[11]Laboratoire des Sciences du Climat et de l'Environnement (LSCE), Commissariat Energie Atomique et aux énergies alternatives - CNRS - Université de Versailles Saint-Quentin-en-Yvelines, 91191 Gif sur Yvette Cedex, France
[12]Université Clermont Auvergne, CNRS, Sigma-Clermont, Institut de Chimie de Clermont-Ferrand F-63000 Clermont-Ferrand, France
[a]Now at Laboratoire de Recherche en Géosciences et Energies (LaRGE), Université des Antilles, Pointe-à-Pitre, Guadeloupe, France
[b]Now at School of Chemical and Pharmaceutical Sciences, Technological University Dublin, Ireland
[c]Now at ATMO Occitanie, Pérols, France

**Correspondence:** Valentin Duflot (valentin.duflot@univ-reunion.fr)

**Abstract.** The Forests gAses aeRosols Clouds Exploratory (FARCE) campaign was conducted in March-April 2015 on the tropical island of La Réunion. For the first time, several scientific teams from different disciplines collaborated to provide reference measurements and characterization of La Réunion vegetation, (biogenic) volatile organic compounds (BVOCs), (bio)aerosols and composition of clouds, with a strong focus on the Maïdo mount slope area. The main observations obtained during this two-month intensive field campaign are summarized. They include characterizations of forest structure, concentrations of VOCs and precursors emitted by forests, aerosol loading and optical properties in the planetary boundary layer (PBL), formation of new particles by nucleation of gas-phase precursors, ice nucleating particles concentrations, and biological loading in both cloud-free and cloudy conditions. Simulations and measurements confirm that the Maïdo Observatory lies within



the PBL from late morning to late evening and that, when in the PBL, the main primary sources impacting the Maïdo Observatory are from marine origin via the Indian Ocean and from biogenic origin through the dense forest cover. They also show that i) the marine source prevails less and less while reaching the Observatory, ii) when in the PBL, depending on the localization of a horizontal windshear, the Maïdo Observatory can be affected by air masses coming directly from the ocean and passing over

the Maïdo mount slope, or coming from inland, iii) bioaerosols can be observed in both cloud-free and cloudy conditions at the Maïdo Observatory, iv) BVOCs emissions by the forest covering the Maïdo mount slope can be transported upslope within clouds and are a potential way of secondary organic aerosols formation in aqueous phase at the Maïdo Observatory, v) the simulation of dynamics parameters, emitted BVOCs and clouds life cycle in the Meso-NH model are realistic, and more advanced Meso-NH simulations should use an increased horizontal resolution (100 m) to better take into account the orography and

improve the simulation of the windshear front zone within which lies the Maïdo Observatory. The FARCE campaign provides a unique set of multi-disciplinary data and results that can be used to better understand the forest-gases-aerosols-clouds system in an insular tropical environment.

## 1   Introduction

Forests, gases - especially biogenic volatile organic compounds (BVOCs) -, aerosols - especially secondary organic aerosols

(SOA) -, and clouds are linked through deep, complex, and interdependent bio-physico-chemical mechanisms. Forests emit BVOCs whose oxidative products may undergo a phase transition and form SOA in the gas-phase chemistry (van Donkelaar et al., 2007; Carlton et al., 2009) or after dissolution and photo-oxidation in cloud water (Liu et al., 2009; Chen et al., 2007; Hallquist et al., 2009; Ervens et al., 2011; Couvidat et al., 2013). Aside from modifying the Earth's radiative budget through their direct effects via scattering and absorption of radiation, aerosols also produce semi-indirect and indirect effects that modify

the dynamics of the atmosphere and the cloud life cycle by adjusting their radiative and microphysical properties. Plants also release primary biological aerosol particles (PBAP) (Després et al., 2012), which have the unique potential to act as cloud- and ice-nuclei under atmospheric conditions that do not trigger these impacts with any other materials (Pope, 2010; Morris et al., 2014). Finally, clouds bring water to forests, some of these latter - the cloud forests - being strongly linked to the regular cycles of cloud formation.

The unrefined mechanisms stated above only exemplify the intricacy of the exchanges and interactions between forests, gases, aerosols, and clouds, which imply microphysics, biochemistry, and dynamics, and remain one of the largest source of uncertainties in the climate system (Kulmala et al., 2004; Fuzzi et al., 2006; Boucher et al., 2013; Gettelman, 2015).

    In a given thermodynamic environment, aerosol particles can be activated and nucleate primary cloud particles. Properties of aerosols related to their role of condensation nuclei (number, size distribution, chemical composition) have a strong impact

on the chemical and physical properties of the formed cloud droplets and widely impact the cloud lifecycle (Twomey, 1977; Albrecht, 1989). The partial absorption of the incoming solar radiation by light-absorbing aerosols inside the cloud locally heats the atmosphere, which can inhibit cloud formation and lead to partial evaporation of the existing clouds. However, the





aerosol effects on clouds remain uncertain due to the complex dynamics-microphysics interactions (Stevens and Feingold, 2009; Boucher et al., 2013).

PBAP are a subset of atmospheric aerosol arising from primary emissions of biological particles including plant and insect debris, spores, pollens, cells, viruses, bacteria and their component parts (Després et al., 2012). PBAP can account for a

substantial fraction (10-90%) of supermicron aerosol number in a wide variety of environments (Pöschl et al., 2010; Huffman et al., 2013; Perring et al., 2015) and they are known to cause diseases and allergenic reactions in humans, animals and plants. Aside from their impacts on global biodiversity and disease transmission, PBAP can potentially affect cloud microphysical processes because of the ability of some PBAP to act as "giant" cloud condensation nuclei (CCN) and ice nuclei (IN) at temperatures up to -2°C (Schnell and Vali, 1975 ; Diehl et al., 2001, 2002 ; Després et al., 2012). The effects of PBAP

on clouds and precipitation processes may strongly affect the location, amount and type (snow vs rain) of precipitation and will have related impacts on cloud lifetime and optical properties (Vali et al., 1976). In part because few measurements exist to constrain their concentrations on regional and global scales, PBAP abundance in the atmosphere is poorly constrained and potential feedback on cloud hydrological pathways is not yet included in climate models (Gabey et al., 2010; Fröhlich-Nowoisky et al., 2016 ; Després et al., 2012). Further, ice-active bacteria and other PBAP have been hypothesized to play

important roles in promoting increased rainfall, regulated by complex natural cycles (Pöschl et al., 2010; Morris et al., 2014, 2017).

Plants contain a number of BVOCs, including isoprene, terpenes, alcohols, aldehydes, ketones and esters (Meigh, 1955), which may be widely distributed throughout plant organs. Individual plant species have unique combinations of these compounds ; consequently, the emission pattern for each species is also specific. For a given plant, its conditions, developmental

stage, the occurrence of injury or damage, and its environment - mainly ambient temperature, light intensity, and air pollution - control BVOCS emissions (Kesselmeier and Staudt, 1999 ; Zemankova et al., 2010). The biological importance of these compounds is still debated ; however, it appears that they may be important in intraplant competition (Harborne, 1988), light or heat damage prevention (Singaas et al., 1997), and defense against pathogens, parasites, or herbivores (Holopainen, 2004).

Because BVOCs substantially impact the tropospheric budget of carbon monoxide (CO), hydroxyl radical (OH) and low-

level ozone (Granier et al., 2000; Poisson et al., 2000; Pfister et al., 2008), they are also an important atmospheric constituent influencing the oxidative capacity of the atmosphere (Houweling et al., 1998; Taraborrelli et al., 2012) on regional (e.g. Solmon et al., 2004; Curci et al., 2009; Sartelet et al., 2012) as well as global scales (e.g. Brasseur et al., 1998; Gauss et al., 2006; Sindelarova et al., 2014).

In addition to the significance in the gas-phase chemistry, SOA can be formed from oxidative products of some BVOCs

through different processes such as homogeneous nucleation, adsorption or absorption (Hyvïrinene et al., 2004; Brégonzio-Rozier et al., 2015), heterogeneous chemistry on preexisting particles or oligomerisation (Kalberer et al., 2004). However, these oxidative products can also be dissolved in cloud water by mass transfer (van Pinxteren et al., 2005) where they can be photo-oxidized and lead to the formation of semivolatile organic compounds (Liu et al., 2009; Mouchel-Vallon et al., 2017). Several studies evaluated the formation of SOAs through the condensation of such semivolatile organic compounds resulting

from cloud evaporation and/or through the reactivity in the aerosols aqueous phase (Chen et al., 2007 ; Hallquist et al., 2009



; Couvidat et al., 2013). Each instant, clouds, which cover 60% of the Earth surface, continuously appear and vanish through evaporation-condensation cycles (Pruppacher and Klett, 1997). Whereas only 10% of clouds precipitate, the remaining 90% evaporate, leading to the evaporation of volatile chemical species and condensation of semivolatile organic compounds on residual aerosol particles. This recently discovered process of SOA formation in clouds is potentially important and still poorly

understood and quantified, as shown by field and lab experiments (Ervens et al., 2010 ; Lee et al., 2012; Pratt et al., 2013).

Clouds are a complex, abstruse, changing manifestation of their environment. Multiphase chemistry can occur within cloud droplets, requiring separate study of each phase, analysis of the interactions at the interfaces, and an understanding of chemical consequences inside each phase. Reactivity in clouds is considerably impacted by the microphysical processes (condensation/evaporation, drops collision/coalescence, freezing/fusion, etc.), which partition the bio-chemical compounds among the

various cloud phases and modify the efficiency of the mass transfer. Photochemical processes inside clouds, and especially inside drops, are strongly amplified with respect to clear sky situations, and homogeneous chemical reactions in the aqueous phase are most of the time faster than the equivalent ones in gas phase (Herrmann et al., 2015). Moreover, the presence of microorganisms in the cloud aqueous phase leads to the biotransformation of chemical compounds and can compete with photo-oxidation mecanisms (Deguillaume et al., 2008, 2014; Vaïtilingom et al., 2010, 2011, 2013). Aside from improving our

knowledge on clouds microphysics, it is therefore crucial to also study the chemical and biological composition and reactivity of chemical species in clouds in order to understand and quantify their efficiency in transforming chemical compounds and altering atmospheric chemistry (gas and aerosol particles).

La Réunion (21°S, 55°E), is a small tropical island located in the south-west Indian Ocean, affected by southeasterly trade winds near the ground, and westerlies in the free troposphere. In spite of transformation of its habitats (Strasberg et al., 2005),

the island still shelters 100,000 ha of native ecosystems, included in a National Park. La Réunion is far from large anthropogenic sources. Thus disconnected from polluted air masses, the island is ideal to study the local processes of chemical formation and transformation of the natural aerosols.

The Maïdo mount is a 2200 m-high summit on the western part of the island. Recent studies of the local circulation in La Réunion (Lesouëf et al., 2011, 2013; Guilpart et al., 2017; Foucart et al., 2018) show that the Maïdo mount is directly under the

influence of air masses coming from the west-north-west downhill slope, which is a zone partially covered by a dense tropical forest. Main primary sources surrounding the Maïdo mount site are therefore from marine origin via the Indian Ocean and from biogenic origin through this dense forest cover. The latter, together with large solar fluxes due to the tropical location and a surface temperature around 20-30°C, is very propitious to BVOCs emission. Moreover, the Maïdo mount site is daily flooded by clouds: they appear around noon and vanish by the evening. Their formation mode (convection on slope) indicates mid-day

interactions between cloud microphysics, BVOCs, SOA and marine aerosols.

A 2160 m-high atmospheric facility was built in 2012 at the summit of the Maïdo mount (Baray et al., 2013). Being inside the boundary layer during the day and frequently near the free troposphere during the night, the Maïdo observatory is dedicated to the study of the low-middle atmosphere (especially in the framework of the Network for Detection of Atmosphere Composition Change - NDACC), as well as to the investigation of the boundary layer composition and processes (especially in the framework

of the Global Atmospheric Watch network - GAW). Several instruments dedicated to the monitoring of the in situ atmospheric



composition (for both gases and aerosols) are permanently deployed at the Maïdo observatory. This facility, also thanks to west-north-west downhill slope of the Maïdo mount linking the Indian Ocean to the facility via dense tropical forest, is therefore a remarkable laboratory for the study of the forests-gases-aerosols-clouds interactions as well as the cloud life cycle in the tropics, from their formation associated with the CCN aerosol properties, to their dissipation through evaporation. Moreover,
the Maïdo observatory is an ideal location to improve constraints on PBAP type and concentrations in a southern hemispheric tropical marine atmosphere.

The importance of the issues presented in the preceding paragraphs, together with the adequacy of the Maïdo mount's slopes and atmospheric facility to study them, prompted the organization of an intensive campaign in March-April 2015: the Réunion FARCE (Forests gAses aeRosols Clouds Exploratory) campaign. Several scientific teams collaborated to provide reference
measurements and characterization of La Réunion vegetation, (B)VOCs, (bio)aerosols and chemical and biological composition of clouds, with a strong focus on the Maïdo mount slope area. This exploratory campaign was thought as a reference socle for projects and campaigns focusing on related matters, specifically the ongoing OCTAVE 2017-2019 (Oxygenated Compounds in the Tropical Atmosphere: Variability and Exchanges) and the starting Biomaïdo 2019-2021 projects. The OCTAVE project aims at studying sources and seasonal cycles of VOC and halogens from forests and Indian Ocean as well as quantify-
ing the contribution of marine/biogenic sources to submicron organic aerosols at Maïdo. The OCTAVE related campaign took place in La Réunion in February-April 2018. The Biomaïdo project focuses on the contribution of the aqueous reactivity to the SOA budget and the related campaign is ongoing (March-April 2019).

The present publication intends to describe the Réunion FARCE campaign set up, and to provide the preliminary results for forest structure, concentrations of VOCs and precursors emitted by forests, aerosol loading and optical properties in the
planetary boundary layer, formation of new particles by nucleation of gas-phase precursors, ice nucleating particles concentration, and biological loading in both cloud-free and cloudy conditions. More detailed and specific results will be published elsewhere. Section 2 gives an overview of the campaign as well as a description of the measurement sites. Section 3 presents the used instruments, methods and model. We finally expose and discuss the preliminary results from this campaign in Section 4.

## 2  Campaign overview and sites description

The FARCE campaign took place March 6 to April 21, 2015 and focused on the area between the coast and the Maïdo observatory (Figure 1). Within the area, different landscape units organized in elevation bands follow each other along the elevation profile. From seashore to 200 m above sea level (asl), human population density is maximal, urban areas are separated by dry grasslands, more or less woody savannah depending on the level of invasion by woody exotic species and orchards. In
the following elevation band up to 900 m asl, sugar cane crops coexist with mixed exotic shrublands and forests present in more slopy areas. The next band comprises pastures, and more generally agricultural croplands up to 1500 m asl. In this band, broad-leaved forests occur in valleys and gullies. Above that band, native broad-leaved forests with various degrees of invasion by exotic plants occupy most of the space. The highest part of this band, up to 1900 m asl, consists in native forests





dominated by the endemic tree species *Acacia heterophylla* (*Fabaceae*) that makes most of standing biomass with individuals reaching 12 to 15 meters high. From place to place, plantations of the coniferous species *Cryptomeria japonica* (*Taxodiacae*) occur with limited extension. Higher up along the elevation profile, the *Acacia heterophylla* forest, locally called "Tamarinaie", transition more or less abruptly with moutain shrublands and heatlands, mostly dominated by endemic shrub species such

as *Erica reunionensis* (*Ericaceae*) and *Stoebe passerinoides* (*Asteraceae*). Field measurements were made at 4 sites within this area (Figure 1): the Cryptomerias plot (21.08°S, 55.34°E, 1248m asl), Chez Henri (21.05°S, 55.35°E, 1305m asl), the Tamarins plot (21.07°S, 55.36°E, 1750m asl), and the Maïdo Observatory (21.08°S, 55.38°E, 2160m asl). Moreover, mobile measurements were performed with a lidar installed on a pick up driving along the slope going from the shore up to the Maïdo Observatory.

To complete the characterization of the La Réunion's forests as well as the related VOCs concentrations, measurements were also made at 4 additional sites distant from the Maïdo mount: one plot was set up in the Bélouve mountain cloud forest in the center of the island (21.06°S, 55.54°E, 1520 m asl), and three plots in the Mare Longue forests in the South-Eastern part (21.35°S, 55.74°E, 150 m, 330 m and 550 m asl) (Figure 1, see Shang et al., 2016 for plot details). The central Bélouve forest is a typical moutain cloud forest occuring on the foothills of the Piton des Neiges extinct volcano between 1100 and 1700 m asl.

The forest lies in a relatively smooth landscape with eastward gentle slope and small gullies incised in the lava substrate. It is daily inundated by clouds and fog flowing from the windward side of the island. Fern and bryophytes epiphytic species, that are dependent on high air relative humidity, are abundant in terms of both species richness and biomass, playing an important role in the local water cycle (Ah-Peng et al., 2017). The forest also hosts a high diversity of woody species with a large dominance of *Acacia heterophylla* however. The three plots in the Mare Longue forest spread along one of the two ecological corridors

on the island, that are uninterrupted native ecosystems in varying conservation state from lowland to highland habitats. The tropical rain forest of Mare Longue makes the lower end of the corridor located South-East of the island on the slopes of the Piton de La Fournaise volcano (the second one being on the drier north-western leeward side of the island). It is the last lowland forest remnant in the Mascarenes. The forest receives several meters of rain each year and is located on recent lava flows of few hundred years of age. It hosts a large diversity of shrub and tree species that form the canopy between 15 and 25 meters

high.

Finally, VOCs concentrations were also measured at two locations with very low vegetation density, both in the volcano area: the Plaine des Sables and Piton de la Fournaise sites (21.23°S, 55.65°E, 2265 m and 21.22°S, 55.68°E, 2360 m, respectively).

In the following, "forest plots" will refer to the Cryptomerias, Tamarins, Bélouve, and Mare Longue (150, 330, 550 m) plots. Note that each of these forest plots has a surface ≈ 0.2 ha.

Table 1 gives a summary of the performed measurements for each site and date.



## 3  Methods, measurements and model

### 3.1  Forest plots characterization

Species, forest structure and Leaf Area Index ($LAI$) are key parameters for forest ecosystem comprehension and modeling. The identification of species provides insight on the related biogenic emissions. The characterization of the forest structure (diameter and height) gives information on the carbon storage capacity. $LAI$ is important for vegetation growth estimation and characterizes the forest interaction surface and exchange efficiency with the atmosphere.

All six forest plots (i.e. Cryptomerias, Tamarins, Bélouve, and Mare Longue 150, 330, 550 m) were surveyed in order to identify species and characterize tree size distribution. Within each plot, all trees with diameter at breast height (DBH) $\geq 7$ cm were identified at species level and localized using cartesian coordinates at plot scale. DBH was measured using rubber tapes, and tree top height (TTH, height of the highest visible tree leaf) was estimated using a lasermeter. For trees with multiple stems, all stems with DBH = 7 cm were recorded, but only the stem with maximal diameter is considered in allometry analyses.

Three different approaches were used to produce estimates of $LAI$ in the Cryptomerias, Tamarins, Bélouve, and Mare Longue 550 m plots. First, hemispherical photographs (hemiphots) were taken in 16 randomly chosen locations during periods without direct sun light reaching the plots. Hemiphots were analysed with Gap Light Analyser software from which the local $LAI$ could be obtained. Three exposure levels were tested with two replicate photographs taken at each location for each exposure level. In a second approach, the LAI2000 plant canopy analyser (Li-Cor, Lincoln, NE, USA) was used to obtain estimates of $LAI$. We estimated $LAI$ at five randomly chosen locations within the plots. The third method used to estimate $LAI$ was based on the vertical decrease in light intensity due to canopy interception described by the adapted Beer-Lambert law (Cournac et al., 2002):

$$I = I_0 e^{-kLAI} \tag{1}$$

where $I_0$ is the incoming light intensity on canopy top measured in open conditions (in clearing or open area), $I$ is the measured light intensity in situ, $LAI$ is the Leaf Area Index above measure point, and $k$ is an extinction coefficient. Here we measured light intensity using two different devices: a luxmeter (LX-1108, Voltcraft, Germany) and a lightmeter (LI-250, LiCor) equiped with a LI-190 quantum sensor for Photosynthetically Active Radiation measurement (PAR). Measures were made sequentially in both open and forest conditions under sky conditions as homogeneous as possible with respect to cloud cover.

### 3.2  BVOCs and precursors

(B)VOCs, nitrogen oxides (NOx), ozone and formaldehyde (HCHO) are deeply linked within the boundary layer, especially in tropical forests (Stickler et al., 2007; Ganzeveld et al., 2008). In the boundary layer, NOx are mainly emitted by fossil fuel combustion and soil, and their main sink is the reaction with OH radical. Boundary layer ozone has several sources, including photochemical production from anthropogenic and natural precursors, the most important being NOx, hydrocarbons and carbon



monoxide (CO). HCHO is directly emitted in the atmosphere (traffic and industrial emissions (Carlier et al., 1986), biomass burning (Lee et al, 1997)), or formed secondary as a result of photochemical reactions. Secondary production of HCHO is initiated in the continental boundary layer by the oxidation of (B)VOCs (Fried et al., 1997).

### 3.2.1 BVOCs

BVOCs were studied at different locations: Maïdo Observatory (2160m) with semi-continuous measurement (12 March - 9 April 2015), Tamarins forest (ground level and on a 10m mast), Cryptomeria forest (ground level), Primary forest (Bélouve) (ground level and on a 10m mast), Mare Longue forest (150 m) and Chez Henri (10m mast) (cf. Table 1).

Active sampling on sorbent cartridges, using a sampling module developed by TERA Environment - a SASS (Smart Automatic Sampling System) - was performed at the Maïdo Observatory. Gaseous compounds were sampled at approximately 10 m

above ground level, using a Teflon sampling line, and then trapped into one multisorbent cartridge, composed of a mixture of Tenax TA 60-80 mesh (250 mg) and carbosieve sIII (150 mg) at 100 mL.min$^{-1}$ during 2 hours. This type of cartridges allowed C4-C14 aromatic compounds, n-alkanes, monoterpenes, isoprene, halogenated compounds to be sampled, at a flow rate of 100 mL.min$^{-1}$. Prior to the sampling, multisorbent filled cartridges were conditioned by flowing purified air through them, at a flow of 30 mL.min$^{-1}$, during 4 hours at 300 °C.

The sampling in the different forest plots were performed on the same cartridges using a manual pump (ACCURO 2000, DRAEGER), and 3 to 6 L of volume sampling at the ground level, and pump (KNC) associated with a mass flow controller for the mast sampling at 10 m.

The analysis of the cartridges were performed using a gas chromatograph - mass spectrometer system (GC/MS, Perkin Elmer) connected to an automatic thermal desorption. Each cartridge was desorbed at 270 °C during 15 min at a flow rate of 40

mL.min$^{-1}$ and reconcentrated on a second trap, at -10°C containing Tenax TA. After the cryofocussing, the trap was rapidly heated to 300°C and the target compounds were flushed into the GC (Keita et al., 2018). The separating column was a 60 m x 0.25 mm x 0.25 $\mu$m PE-5MS (5% phényl, 95% PDMS) capillary column (Perkin Elmer). The temperature profile of the GC was ramped (35°C for 5 min, heating at 8°C.min$^{-1}$ to 250°C, hold 2 min). The chromatography parameters were optimized to enable good separation of circa 80 identified compounds; a complete run took about 34 minutes. The mass spectrometer was

operated in Total Ion Current (TIC) from 35 to 350 m/z amu.

### 3.2.2 NOx, ozone and HCHO at Maïdo Observatory

At the Maïdo Observatory, NO$_x$ (=NO+NO$_2$) were monitored with an Environnement SA AC31M using an ozone chemiluminescence technique. Detection limit is 0.35 ppbv and 12% of uncertainty.

Ozone was monitored with a UV photometric analyser (Thermo Scientific model 49i). Model 49i uses a dual-cell photometer

and measures amount of ozone in the air from 0.05 ppb concentrations up to 200 ppm with a response time of 20 seconds and a precision of 1 ppb.

Formaldehyde (CH$_2$O) measurements were made with a modified Aero-Laser AL4001, a commercially available instrument. This instrument is based on the Hantzsch technique which is a sensitive chemical fluorimetric method that is specific to CH$_2$O.



The transfer of formaldehyde from the gas phase into the liquid phase is accomplished quantitatively by stripping the $CH_2O$ from the air in a stripping coil with a well defined exchange time between gas and liquid phase. Formaldehyde was measured at thirty seconds time intervals and the detection limit was 100 pptv. A full description of the instrument and its performance is given in Junkermann and Burger (2006).

## 3.3 Profiling of boundary layer aerosol optical properties

We used a mobile aerosols lidar in synergy with a handheld sunphotometer to retrieve optical properties of the encountered aerosols in the planetary boundary layer (PBL) along the Maïdo mount slope during 3 days (Table 1). The lidar system used in this study is a LEOSPHERE ALS450 based on a Nd:Yag laser, producing pulses with a mean energy of 16 mJ at 355 nm and a frequency of 20 Hz. Lidar measurements have been averaged over 2 min with a vertical resolution of 15 m. The lidar profiles enable us to retrieve atmospheric structures (boundary layer heights, aerosol layers and clouds) and optical properties (lidar ratio (LR) and extinction coefficient) in synergy with sunphotometer measurements. It is particularly well-adapted to the study of the PBL thanks to its full-overlap height reached at ≈150 m. A more complete description of the lidar and its instrumental features can be found in Duflot et al. (2011). For this campaign, the system was installed in an air-conditioned box adapted for use in severe conditions and attached on a pick up platform to perform mobile observations. The energy was supplied by 6 batteries connected to a power inverter, which give a ≈3 hours autonomy to the system.

Aerosol optical thickness ($AOT$) measurements were performed in clear-sky conditions using a MICROTOPS II Sun photometer instrument (Solar Light, Inc.). The instrument field of view is about $1°$. The $AOT$ is measured at five wavelengths (440, 500, 675, 870 and 1020 nm). The instrument was calibrated at the NASA Goddard Space Flight Center against the AERONET reference CIMEL Sun/sky radiometer. The data presented here have been quality- and cloud- screened following the methodology of Smirnov et al. (2000) and Knobelspiesse et al. (2003) and the mean uncertainty on the $AOT$ measurements equals 0.015 (Pietras et al., 2002). The $AOT$ at the lidar wavelength of 355 nm ($AOT_{355}$) was calculated from $AOT_{440}$ using the Ångström exponent (Ångström, 1964) between 440 and 675 nm. The uncertainty on the retrieved $AOT_{355}$ has been computed following the approach showed by Hamonou et al. (1999).

The synergetic approach between lidar and sun photometer measurements to calibrate the lidar system, to retrieve aerosols optical properties, and to evaluate the uncertainties can be found in, e.g., Duflot et al. (2011). It is noteworthy that this method gives access to a height-independent LR value. The aerosols optical properties given hereafter are therefore valid for the mixture of aerosols encountered above the lidar at one point on the trajectory of its carrier.

## 3.4 Aerosols size distribution

Aerosols were characterized at the Maïdo Observatory site (cf. Table 1) for their size distribution using a Differential Mobility Particle Sizer (DMPS) in the 10-550 nm size range. The DMPS was custom-built with a TSI-type Differential Mobility Analyzer (DMA) operating in a closed loop and a Condensation Particle Counter (CPC, TSI model 3010). Particles were charged to equilibrium using an Ni-63 bipolar charger at 95 MBq. The quality of the DMPS measurements was checked for flow rates and RH according to the ACTRIS recommendations (Wiedensohler et al., 2012). DMPS measurements were performed down




a Whole Air Inlet with a higher size cut-off of 25 $\mu$ m (under average wind speed conditions of 4 m.s$^{-1}$) (Tulet et al., 2017 ; Foucart et al., 2018).

## 3.5 Cloud-free size-resolved ice nucleus concentrations

Measurements of ice nucleating particle (INP) concentrations in the immersion mode were made at the Maïdo Observatory site (cf. Table 1). To estimate INP number, the micro-orifice uniform deposit impactor-droplet freezing technique (MOUDI-DFT) was used to collect and analyze INP on a size-resolved basis between 0.18 and 18 $\mu$m (Mason et al., 2015). Each aerosol was impacted onto glass microscope cover slips coated in a hygroscopic material using a MOUDI operated at a constant flow rate of 30 L.min$^{-1}$. Each stage of the MOUDI cascade impactor collects a discreet range of particle sizes, and so subsequent

analyses provide information about INP as a function of aerodynamic particle diameter. A total of 23 samples were collected for 6-7 hours each and then analyzed using the DFT technique. Briefly, humidified air is passed over each sample until liquid droplets form around deposited particles. The temperature of the stage is reduced at a rate of -10°C.min$^{-1}$ from 0°C to -40°C while images of the stage are recorded by a video camera. Post-analysis records the number and size of droplets that freeze as a function of temperature, and these data are converted to INP concentrations as a function of both particle size and freezing

temperature.

## 3.6 Cloud-free fluorescent particle counts

The Wide-band Integrated Bioaerosol Sensor (WIBS) was installed at the Maïdo Observatory during the study period (cf. Table 1). This instrument measures size and fluorescence from individual particles from ≈0.8 to 10 $\mu$m. Particles are exposed to intense flashes of UV light at 280 and 370 nm and the resulting fluorescence is imaged onto two photomultiplier tubes

filtered to detect light from 310-400 nm (saturated by the 370 nm flash) and from 420-650 nm. Each particle therefore may have signal above background in any of three channels, denoted here as channels A (fluorescence detected between 310 and 400 nm following 280 nm excitation), B (fluorescence detected between 420 and 650 nm following 280 nm excitation) and C (fluorescence detected between 420 and 650 nm following 370 nm excitation). The excitation wavelengths and detection bands are chosen to correspond roughly to those of tryptophan and nicotinamide adenine dinucleotide (NADH), two compounds

commonly found in biological systems, although other fluorophores likely also contribute to the fluorescent activity of airborne microbes.

Concentrations of fluorescent particles are then used as a proxy for PBAP. The WIBS can distinguish specific classes of PBAP (i.e. bacteria, spores and pollen) from one another using the measured size and fluorescent intensities and following an analysis framework published by Perring et al. (2015) and referenced to published responses to known bioaerosol samples

(Hernandez et al., 2016 ; Savage et al., 2017). In this framework each particle is assigned a unique "type" indicating which, if any, of the three possible channels had signal above background for that particle. Particles that have fluorescence above background in a single channel are assigned as type A, B or C while particles with signal above background in multiple channels are led with each relevant channel (i.e. type AB particles have signal above background in channels A and B but not in C while type ABC had signal above background in all channels) and particles with no detected fluorescence in any




channel are designated "non-fluorescent". Application of this framework in laboratory studies shows that both bacteria and anthropogenic combustion sources tend to be detected as type A at small ($\approx$1 $\mu$m or less) while fungal spores appear as a combination of types A, AB and ABC at slightly larger sizes (2-5$\mu$m) and pollen is detected as a combination of C, BC and

ABC with less size specificity due to the prevalence of pollen fractionation. Very little that is biological appears as types B or AC; high concentrations of type B have been associated with biomass burning aerosol (Savage et al., 2017) and type AC is very rarely observed in either laboratory or ambient studies.

### 3.7 Biological composition of clouds

Chemical and microbial characterization was performed on 5 cloud water samples collected at La Réunion during the FARCE
campaign (cf. Table 1). A mobile sampler was deployed during the campaign to sample and characterize clouds. A cloud sampler together with weather sensors and cloud droplet probe (CDP) were installed on a 10 m mast to avoid ground contamination. The cloud samples were collected with a single stage cloud collector under sterile conditions suitable for microbial analysis (Kruisz et al., 1993; Vaïtilingom et al., 2012). The chemical composition of the clouds was identified by determining the concentrations of the main inorganic ions and the major carboxylic acids using ionic chromatography method (Jaffrezo et
al., 1998), together with the concentrations of formaldehyde and hydrogen peroxide using fluorimetric methods (Vaïtilingom et al., 2013).

For the microbial analysis, the total microbial cells count has be done using flow cytometry (Hammes et al., 2008). The energetic state of the cloud's microorganisms was determined by measuring the ATP and ADP concentrations using a bioluminescent method (Lundin et al., 1986). The cloud water samples were cultivated in R2A agar medium immediately after
their sampling and incubated 1 week at 20°C to allow the formation of microbial colony-forming. The bacteria-like (typically smooth and circular) and fungi-like (typically filamentous) colonies that formed (CFU) were counted, and then isolated and purified on R2A agar medium at 17°C for further identification by ribosomal RNA gene sequencing using the same protocol as Vaïtilingom et al. (2012). Bacteria and yeast were identified on the basis of their 16S or 26S rRNA gene sequences, respectively.

### 3.8 The Meso-NH model

The mesoscale, non-hydrostatic atmospheric model Meso-NH (Lac et al., 2018) is able to simulate both idealized systems and real meteorological events at high resolution on large domains with complex terrain. A full description of the model capabilities is available at http://mesonh.aero.obs-mip.fr/.

The model is set up with two two-way nested domains with horizontal grid spacing of 2 (D1) and 0.5 (D2) km and grid sizes of 128 X 128 and 200 X 180 points, respectively. The innermost domain is centered over La Réunion. In the vertical, 64 levels
are used, with highest resolution near the surface. ECMWF analysis were used to initialize the model on 1$^{st}$ April 2015 at 12:00 UTC, and to feed the lateral boundaries conditions. In the two domains, the deep convection is explicitly resolved. In the outermost domain, the shallow convection is parameterized following Pergaud et al. (2009). The microphysics scheme (Pinty and Jabouille, 1998) is a single-moment bulk scheme that predicts the mixing ratio of five microphysical species: cloud water, rain, cloud ice, snow and graupel. This scheme is derived from Lin et al. (1983). For both domains, the turbulence scheme



was set in a 3D mode with the 3D mixing length of Deardorff (1980) used. The radiative scheme is the one used at ECMWF (Gregory et al., 2000) including the Rapid Radiative Transfer Model (RRTM) parameterization (Mlawer et al., 1997). The chemical parameterization are upon Suhre et al. (1998) and Tulet et al. (2003, 2006) (gas chemistry). The biogenics emissions

are from the version 2.1 of the Model of Emissions of Gases and Aerosols from Nature (MEGAN, Guenther et al., 2012; Sindelarova et al., 2014) computed on-line by the SURFEX scheme (https://www.umr-cnrm.fr/surfex/).

## 4    Preliminary results and discussion

### 4.1    Forest plots characterization

We surveyed all six forest plots in order to characterize forest structure, whereas we could implement methods for $LAI^*$ in

four plots only (cf. Table 1). Overall, 56 different species were surveyed in the plots with various species richness (Table 2). The most diverse plots are located in the Mare Longue rainforest area, with a maximum of 34 species at 550 m asl. These plots also exhibit high basal area in relation with high maximal DBH and height. The two *Acacia*-dominated plots (Bélouve, BELO and Tamarins, TAMA) showed low woody species diversity, with two species only present in TAMA (*Acacia heterophylla* and *Erica arborescens*). They showed lower basal area (sum of tree trunks sections) and lower tree size compared to the lowland

plots in Mare Longue forest. The low basal area and the relatively low number of trees in the Bélouve forest plot (BELO) could indicate former unknown logging of *Acacia heterophylla*. Last, the forest plot set up in the plantation of the exotic species *Cryptomeria japonica* (Cryptomerias, CRYP) hosts two species, one present as one individual only (*Eucalyptus robusta*).

The analysis of tree diameter and height distribution (Figure 2) reveals highly different structures, from the homogeneous and simple structure of the plantation plot (CRYP), the bi-layered structure of *Acacia heterophylla* forests (BELO and TAMA)

as shown by the bimodal distributions of both size variables, to the more heterogeneous structure of the lowland rainforest in Mare Longue plots, as shown by higher maximal sizes and more even distributions. This variety of forest structure is also well represented in allometric relationships between tree height and diameter differing across plots (Figure 3). These relationships highlight the role of particular species in forest structure. Hence, the lower layer in the Bélouve forest (BELO, Figure 3) is mostly composed of the tree-ferns *Cyathea borbonica* and *Cyathea glauca*, of *Erica arborescens* in the Maïdo forest (TAMA),

and by *Pandanus purpurascens* in the plot of Mare Longue forest at 550 m asl (MALO550).

Regarding $LAI$, for the light interception method, we present here estimated values of $LAI^* = kLAI = exp\left(\frac{I}{I_0}\right)$, because we did not have independent estimates of the extinction coefficient $k$. The four tested methods provided roughly similar estimates (Figure 4), with however changing ranks across plots, except for hemiphots in MALO550 which under-estimated $LAI^*$, and the luxmeter method in TAMA which over-estimated $LAI^*$ in TAMA. The highest values of $LAI^*$ were found in

the more rainy site of MALO550 which is consistent with the high biomass present in the plot, as shown by basal area and stem number (Table 2). The value obtained for the BELO plot ($< 1$) is surprising as it is more typical of much drier ecosystems with low biomass and foliage density. It is however consistent with the evidenced low stand density, that seems to indicate former logging. Nevertheless, it is also close to the estimate found for the TAMA plot ($\approx 1.5$) located in *Acacia heterophylla* dominated stands. This species is well-known for displaying leaf-shaped petioles (phyllodes), photosyntically active, but few





relative to tree size, and for its clustered foliage in large crowns, which also may partly explain the deviation of the luxmeter estimate for the TAMA plot.

The extinction coefficient $k$ of the light interception method is related partly to the optical properties of the leaves and mainly

to the structural properties of the canopy (height, stem density, leaf clustering and inclination) (Cournac et al., 2002). It also depends on the radiation waveband that is measured, and $k$ values are typically in the range 0.70 - 0.94 (Wirth et al., 2001). As estimates obtained by the light interception method are relatively close to estimates by hemiphots, our results suggest that in the case of the forests studied here, $k$ is rather close to the higher end of known values, although it certainly varies from one forest type to another.

Shang et al. (2016) measured higher $LAI$ values for the Tamarins, Cryptomeria, Bélouve and Mare-Longue 550 plots using vertical profiles of an airborne LIDAR with a large footprint. Having in mind that the nutrient availability in the ground can vary from one point to another in a a given forest, and consequently so does the $LAI$, these differences between ours and theirs retrieved $LAI$ values are due to the fact that the airborne LIDAR samples a much wider area compared to the measurements performed from the ground in a much smaller plot ($\approx$ 0.2 ha).

The results of these species identification and $LAI^*$ measurements were used to improve the simulation of emitted BVOCs in the Meso-NH model, based on previous studies from Guenther et al. (1995, 2012).

### 4.2   Atmospheric dynamics

Meteorological regimes and dynamics on La Réunion have been widely studied by Lesouëf et al. (2011, 2013), Durand et al. (2014), Guilpart et al. (2017), Tulet et al. (2017) and Foucart et al. (2018). These works show that the island is affected by a

south-easterly trade winds regime, which is intense in winter (June-August) and moderate in summer (December-February). This main weather regime causes an acceleration of the winds around the mountains on the south-west and north-west sides of the island. The north-western sector of the island is screened from the trade winds by high mountains, which allows the development of diurnal thermally induced circulations, combining downslope (catabatic winds) and land breeze at night, and upslope (anabatic winds) and sea breeze during the daytime. Moreover, a returning loop occurs almost every day on the north-

west side of the island in the boundary layer. This north-westerly dynamic flow joins the upslope and sea breeze during the daytime to bring north-western air masses up to the Maïdo planèze. This convection on slope causes a quasi daily formation of clouds, which are usually weakly developed vertically with a low water content.

This process is clearly visible on the Meso-NH simulation of the $2^{nd}$ of April 2015 (Figure 5, left panel). One can see in particular the returning loop on the north-west sector and a related west-north-westerly flow on the west flank of the mountain.

One can also notice a horizontal windshear front located under the Maïdo Observatory (noted M) and the Tamarins plot (noted T). This surface front zone coincides with the confluence of the south-easterly trade wind regime passing over the mountain and the convection on slope north-westerly flow. The location of this frontal zone, which moves along the day, is very important as it drives the origin of the air masses that are sampled at the Maïdo Observatory: in a north-westerly regime, the observatory will sample air masses loaded with species coming from marine, anthropogenic and biogenic sources; in a south-easterly trade wind regime, air masses come mainly from 2000m-altitude and are less impacted by surface emissions.





Figure 6 shows the time series of the wind direction, air temperature, and integrated cloud water content simulated during 4 days by Meso-NH at the surface level at the Maïdo Observatory and Tamarins plot. Observations of wind direction and air temperature at the Maïdo Observatory are also shown (red crosses). At the Maïdo Observatory, nighttime observations

show a clear prevailing easterly flow ; daytime observations confirm that the Observatory is located within the windshear front zone between the trade wind flow (south-easterly) and the upwelling slop thermal breeze (north-westerly) (Figure 5, left panel). Temperatures range from 12 °C at night to 21 °C during daytime. It is worth mentioning that these observations should be taken as general tendencies as the sensors location on the concrete roof of the Observatory makes them sensitive to the warming up of the building as well as to the turbulences generated by the building. Keeping in mind this caveat, and

considering the observations error bars and the model standard deviations, one can see that there is an overall agreement between the measurements and the simulations. However, the change in the wind direction from easterly to southerly observed at Maïdo on the $2^{nd}$ of April at 6 UTC is not captured by the model: the location of the Observatory within the convergence zone makes difficult the numerical simulation of these flows, especially with a 500 m horizontal resolution. This high horizontal variability is clearly shown by the deviations of the simulated values around the model points (vertical bars on Figure 6): the

dispersion of the simulated values is higher during daytime at the Maïdo Observatory than at the Tamarins plot.

At the Tamarins plot, the wind direction simulated by the model is very stable with a westerly flow between 5:00 and 15:00 UTC linked to the upwelling thermal breeze. Outside this period, the flow is easterly. Temperature fluctuations are also regular along the 4 simulated days. Maximum values reach 18-19°C at around 4:00 UTC and temperature remains steady until 11:00 UTC. Nighttime simulated minima are around 16°C.

The simulated cloud life cycle exhibits a clear day-long period. On the Tamarins plot, clouds appear around 8:00 UTC and vanish around 13:00 UTC. The simulated integrated cloud water content is low (between 0.1 and 0.2 mm), which is typical for clouds weakly developed vertically. This clouds formation causes a PAR drop (not shown), which strongly reduces BVOC emissions from local noon (8:00 UTC).

At the Maïdo Observatory, simulated clouds are sporadic, and, when they appear, their water content is low. These results

support the conclusion that the Observatory is located within a cloud vanishing zone, which is linked to the flow shearing off and the easterly transport of dry air.

### 4.3   BVOCs and precursors

The mean measured isoprene mixing ratio (pptv) for all sites is represented on Figure 8. As expected, lowest concentrations of isoprene were found in the volcanic area with a very low vegetation density (<25 pptv), and the maxima of isoprene

concentration were found within the forest plots ($\approx 195 \pm 68$ pptv in the Mare-Longue 150 m plot and $\approx 110 \pm 38$ pptv in the Bélouve and Tamarins plots). Figure 5 (right panel) gives an example of isoprene surface concentration simulated by Meso-NH on the $2^{nd}$ of April at 6:00 UTC. The highest simulated concentrations are located in the main rain forests on the east coast of the island (more dense) and at middle or low altitudes (more favorable to an optimal leaf temperature ; see Guenther et al., 1995). In the north-east and south forests, the model simulates isoprene concentrations of 800 and 400 pptv, respectively. On the west side, concentrations higher than 500 pptv are less extended due to higher urbanisation and lower vegetation density.





On the Maïdo planèze, the maximum simulated isoprene concentration is 400 pptv at 500m asl. Considering the error bars and standard deviations, one can see that there is an overall agreement between the measurements and the simulations at the island scale.

Figure 6 (lower panels) shows the time series of isoprene measurements and simulations at the Maïdo Observatory and at the Tamarins plot from the $2^{nd}$ to the $5^{th}$ of April. At the Maïdo Observatory, observations exhibit a daily cycle of isoprene concentration, with maxima between $50 \pm 17$ and $90 \pm 31$ pptv during daytime. The isoprene concentration of $40 \pm 14$ pptv at 20:00 UTC on the $2^{nd}$ of April (00:00 LT on the $3^{rd}$ of April) is unexpected and would need further investigation. At the Tamarins plot, the isoprene measurements performed along the $2^{nd}$ of April show a similar diurnal cycle with concentrations

higher than the ones at the Maïdo Observatory: $200 \pm 68$ pptv at 04:00 UTC (08:00 LT) and $100 \pm 34$ pptv at 10:00 UTC (14:00 LT). At both sites, the range of simulated isoprene concentrations agree remarkably well with the observations. Simulations exhibit a clear daily cycle with an abrupt increase at sunrise (03:00-04:00 UTC and 07:00-08:00 LT) and decrease toward 0 pptv at sunset. Taking into account error bars and standard deviations, one can see that there is an overall agreement between the measured and simulated time series of isoprene concentrations at both sites. At the Tamarins plot, the simulated isoprene

concentrations along the $2^{nd}$ of April are strikingly well in agreement with the observations. At the Maïdo Observatory, the daytime situation is more complex since the Observatory is located within the windshear front zone between the trade wind flow (south-easterly) and the upwelling slop thermal breeze (north-westerly) (Figure 5, left panel). Simulated isoprene concentrations are directly linked to the direction of the simulated flow within which lies the Maïdo Observatory. On the $3^{rd}$ and $4^{th}$ of April, the simulated north-westerly flow is intense, the Maïdo Observatory lies within an air mass that came up

along the Maïdo mount slope and, consequently, simulated isoprene concentrations reach 200 pptv on the $3^{rd}$ and $4^{th}$ of April. On the $2^{nd}$ and $5^{th}$ of April, the model keeps the Observatory in a south-easterly flow, except between 8:00 and 11:00 UTC on the $2^{nd}$ of April and at 8:00 UTC on the $5^{th}$ of April when the flow is northerly. Maximum simulated isoprene concentrations are therefore lower than on the $3^{rd}$ and $4^{th}$ of April and stay below 80 pptv. On the $2^{nd}$ of April, the observations show that the model underestimates the upwelling thermal breeze and that the Observatory is in a westerly regime during daytime. Inversely,

on the $5^{th}$ of April, observations show that the Observatory is within a continuous easterly flow, whereas the model simulates a situation more similar to the one of the $2^{nd}$ of April. To sum up, Meso-NH seems to simulate very well dynamics and isoprene concentrations at the Tamarins plot, both in terms of range of values and variations. At the Maïdo Observatory, simulations and observations agree very well in terms of range of values, and reasonably well in terms of variations, due to the location of the Observatory within the windshear front zone.

The Meso-NH model is used in this paragraph to provide an insight into the diurnal evolution on the slope of the Maïdo mountain. Figure 7 shows the simulated vertical cross-section of the isoprene concentration along the transect shown on Figure 5 (right panel) on the $2^{nd}$ of April at 6:00 (left panel) and 9:00 UTC (right panel). At 6:00 UTC, one can see a weakly developed boundary layer and an absence of clouds along the slope. With a temperature equal to 19°C at 1700 m (Tamarins plot, see Figure 6) and a stable atmosphere, Meso-NH simulates an accumulation of biogenic compounds close to the surface.

For example, the maximum simulated isoprene concentrations are around 150-200 pptv at 1000 m asl within a 300 m-thick layer. This thickness coincides with the top of the stable layer shown by the $4 \times 10^{-4}$ s$^{-2}$ isoline of the square equivalent



Brunt-Väisälä frequency. At 9:00 UTC, convection develops with an upwelling thermal breeze generating clouds, which base is located around 800 m asl and top around 2000 m asl. These clouds are in contact with the ground between 1000 and 1500 m asl, which is in agreement with visual observations and LIDAR measurements (cloud layer observed from $\approx$ 1700 to $\approx$

2100m at 8:10 UTC ; see Section 4.4 and Figure 12). One can notice on Figure 7 that the zone where clouds are in contact with the ground coincides with the forest areas of the Maïdo planèze. The presence of clouds causes a PAR drop, which bounds isoprene emissions, and the model simulates a decrease of the maximum concentrations down to 100 pptv. Isoprene produced before the formation of clouds is vertically transported by convection and concentrations of 50 pptv are simulated in aqueous phase. The result is important for two reasons. First, this species can be oxydized (Carlton et al., 2007; Liu et al., 2009) or

oligomerised (Renard et al., 2015) within the cloud to form secondary less volatile species, and potentially SOA. This is then a potential way of SOA formation at the Maïdo Observatory. Second, this species can be used as a proxy for gases and PBAP emitted by the vegetation on the Maïdo slope.

Consequently, these preliminary results support the conclusions that (i) the simulation of dynamics parameters, emitted BVOCs and clouds life cycle in the Meso-NH model are realistic, and (ii) more advanced Meso-NH simulations should use an

increased horizontal resolution (100 m) to better take into account the orography and improve the simulation of the windshear front zone within which lies the Maïdo Observatory.

Figure 9 shows the diurnal variation of isoprene concentration, temperature and PAR in the Bélouve plot on the $14^{th}$ of April. One can see that the variations of these three parameters are well correlated (except at 16:00 local time when the isoprene concentration increases while the temperature and PAR decrease, which would need further investigation).

At the Maïdo Observatory, HCHO was continuously monitored from the $11^{th}$ of March to the $11^{th}$ of April 2015. We observe a large diurnal variation of HCHO during the whole studied period, with maxima between 8:00 and 10:00 UTC (12:00 and 14:00 LT) reaching up to $\approx$ 4 ppbv (Figure 10). This period of time is associated with higher solar radiation and higher surface air temperatures that promote the occurrence of photochemical reactions (Lelieveld et al., 2002). During the night, the Maïdo observatory is in the free troposphere under the oceanic influence (Baray et al, 2013; Guilpart et al., 2017). The low

nighttime concentration of HCHO (<0.4 ppbv) is representative of the marine unperturbed environment. During daytime, it is influenced by the compounds from the mixing layer.

Biogenic compounds like isoprene react rapidly with OH to form secondary HCHO. The scatter plot of HCHO versus NOx and ozone is shown on Figure 11. Concentration of HCHO is mainly driven by 2 different situations. For low NOx situation (< 2ppbv), high HCHO is correlated with high ozone, both of them are photochemically produced from VOCs photooxidation

including isoprene oxidation. For high NOx situation (> 2ppbv), HCHO is well correlated with NOx and low ozone, which is due to anthropogenic emissions along the coast and the upwind situation.

## 4.4 Boundary layer aerosols along the Maïdo slope

On the $2^{nd}$ of April, the lidar system was operated on the pick up from the shore up to the Maïdo station (Figure 12, top panel). Figure 12 (bottom panel) gives the time series of the lidar backscattered signal along the track of the pick up (blank periods are due to defaults of the system). At the beginning of the track, one can see an aerosol-loaded layer extending up to $\approx$1500m





above ground level (agl), narrowing gradually while the pick up goes up. Clouds start to appear at ≈250m agl when the lidar is at ≈750m asl (≈10:30 local time - LT).

Figure 13 shows the extinction profiles retrieved using simultaneous sun photometer measurements at 9:27, 10:23, 11:32 and
12:14 LT (crosses on top panel and vertical dashed lines on bottom panel of Figure 12). Table 3 summarizes the lidar altitude ($Z_l$) and distance from the shore ($Ds$), the measured $AOT_{355}$ and Ångström coefficient between 500 and 870 nm ($\mathring{A}$), as well as the retrieved LR at 355 nm ($LR_{355}$) and vertical extension of the aerosol layer ($H_a$) for each of these four local times. One can first notice that $AOT_{355}$ decreases with the lidar altitude $Z_l$ (from $0.08 \pm 0.02$ at 40 m to $0.02 \pm 0.005$ at 2160 m), and that, consistently, the retrieved extinction profiles show a decreasing $H_a$ with increasing $Z_l$ (from ≈1400 m at 40 m to ≈800
m at 2160 m), showing that the thickness of the probed aerosol layer decreases while the lidar is getting closer to the free troposphere. Oppositely, $\mathring{A}$ increases with $Z_l$ (from $1.06 \pm 0.16$ at 40 m to $1.38 \pm 0.21$ at 2160 m), showing that the probed aerosols get smaller while the lidar gets further from the shore. Similarly, $LR_{355}$ increases from $30 \pm 8$ sr at 40 m to $50 \pm 13$ sr at 2160 m.

While the $\mathring{A}$ values give information on the average size of the encountered aerosols, $LR_{355}$ values give information on the
absorption efficiency of the particle ensemble, and thus on the type of aerosols. At 9:27 LT, $LR_{355}$ equals $30 \pm 8$ sr, which is in the range of typical $LR_{355}$ values for marine aerosols (sea salts) (Cattral et al., 2004; Duflot et al., 2011). At 10:23 and 11:32 LT, getting further from the shore, $LR_{355}$ rises up to $41 \pm 10$ sr and $50 \pm 13$ sr, respectively, which is higher than typical $LR_{355}$ values for marine aerosols and is most probably a signature of a marine, anthropogenic/urban and biogenic aerosols mixture. To sum up, between the shore and the Maïdo station, the aerosol optical depth, the aerosol layer thickness and the
aerosol average size decrease, while their absorption efficiency increases: the lidar gets closer to the free troposphere and the sea salt loading decreases.

At 12:14 LT (Tamarins plot), $H_a$ and $AOT_{355}$ values (≈850 m and $0.05 \pm 0.01$, respectively) are slightly higher that the ones at the Maïdo station (≈800 m and $0.03 \pm 0.008$, respectively), showing that the probed aerosol layer is slightly thicker at the Tamarins plot than the one probed at the Maïdo station. $\mathring{A}$ values are similar at the two measurement locations
($1.30 \pm 0.20$ and $1.38 \pm 0.21$ at the Tamarins plot and at the Maïdo station, respectively), which indicates that the average size of the probed aerosols is similar. However, surprisingly, $LR_{355}$ equals $88 \pm 22$ sr at the Tamarins plot, which is in the range of typical $LR_{355}$ values for relatively strongly absorbing aerosols, such as combustion aerosols (biomass burning and anthropogenic/urban/industrial aerosols). Our best case scenario to explain this $LR_{355}$ value is a contribution from local anthropogenic/urban sources reaching the Tamarins plot at this time.

**4.5   Nucleation and bio-aerosols**

The previous sections focus on the general boundary layer dynamics, clouds life cycle, meteorological parameters, BOVCs and their precursors prevailing on La Réunion and, especially, on the Maïdo mount slope during the FARCE campaign. The present section focuses on the impacts of these prevailing conditions on the formation of aerosols, INP, and biological loading in both cloud-free and cloudy conditions.





### 4.5.1    Impact on aerosols formation

The formation of new particles by nucleation of gas-phase precursors was observed every day of the field campaign except on the $30^{th}$ and $31^{st}$ of March and on the $6^{th}$ of April. A typical new particle formation event occurring during the campaign is shown Figure 14 for the $3^{rd}$ of April.

The aerosol concentration is increasing at the lowest size range from 03:00 UTC and progressively at slightly higher diameters during the following hours, showing the aerosol early growth after they have been formed. On the $3^{rd}$ of April, the observed (and modeled) wind direction is shifting from easterly to westerly at 04:00 UTC (Figure 6). In parallel, the isoprene concentration is observed (and modeled) to increase from around 03:00 UTC (Figure 6), in correspondence with the appearance of ultrafine particles. The appearance of isoprene concentrations one hour before the wind shift would correspond to the advection of an eastern counterflow over the Mafate area, that was in contact with the vegetation on the eastern side of the island. Hence, for explaining the presence of ultrafine particles at 03:00 UTC, only biogenic or marine precursors can be invoked.

At around 06:00 UTC, the intensity of the nucleation mode is abruptively increased, followed one hour later by the arrival of an additional mode particles in the 80-100 nm size range (accumulation mode particles). The appearance of such an additional mode indicates that the boundary layer reaches the site, and contains primary particles. The growth of the newly formed particles continues during the day by condensation of low volatility gas-phase compounds, until they reach 40-50 nm. Note that it is not clear which precursors are responsible for the aerosol growth, as marine, biogenic and anthropogenic contributions are expected along the air mass track. Some of these particles may directly contribute to the CCN population for high cloud sursaturations, or they can grow to larger sizes during the following day to be active as CCN at lower sursaturation. At 16:00 UTC, the air mass is changing to a particle-poor regime, indicating that the station is sampling free tropospheric air again, under the influence of catabatic winds going downhill (cf. Section 4.2).

### 4.5.2    Ice nucleating particles concentration

The results of the MOUDI-DFT measurements performed at the Maïdo Observatory are summarized in Table 4. The average INP concentration values observed here are remarkable, in part, because of the low values observed at each temperature. For contrast, INP concentration values summarized by Mason et al. (2016) from six locations in North America and Europe, also determined using the MOUDI-DFT technique, are shown in the table. At the two temperatures (-25°C and -20°C) available for comparison, concentration values reported by Mason et al. (2016) are 1-2 orders of magnitude higher. In addition to the low concentrations, the relative fraction of large particles (>1.0 $\mu$m) observed at the Réunion site are substantially higher than the range of similar size fractions reported by Mason et al. (2016). Because of the long sample periods (6-7 hrs), MOUDI samples were generally mixed from multiple air mass types (i.e. free tropospheric, trade wind, or locally emitted), and so specific conclusions about air mass origin is challenging.

Much has been recently stated about the lack of observations in remote areas of the southern hemisphere. The results summarized here are suggestive that INP concentrations in relatively remote stations such as on La Réunion may indeed be



considerably lower than in areas more commonly accessed for observation in continental areas of the northern hemisphere, as has been suggested previously (e.g. Burrows et al., 2013; McCluskey et al., 2018).

### 4.5.3  Biological loading in cloud-free and cloudy conditions

Figure 15 shows supermicron fluorescent aerosol loadings observed by the WIBS during daytime (11:00 - 18:00 LT) and night-time (midnight to 8:00 LT) for the duration of the project color coded by the fluorescent particle type. Daytime concentrations are generally higher than nighttime concentrations although even the highest daytime concentrations observed are low to moderate in comparison to those reported for various continental locations (Schumacher et al., 2013; Perring et al., 2015). Both the fluorescent particle type distribution (dominated by types A and AB) and observed number distributions (not shown but peaking between 2 and 5 $\mu$m for types A, AB and ABC) are similar to what is observed for laboratory sampling of fungal spores. The contribution from type B particles is sometimes substantial and likely reflects a local biomass burning influence, a hypothesis also supported by the type B number distribution which shows the number distribution increasing down to 0.5 $\mu$m (the lower limit of detection of the WIBS) implying a peak below this size, likely coincident with the accumulation mode. Fluorescent aerosol concentrations were lowest early in the campaign during a strong tropical storm (named "Haliba", which was ≈60 km south-west of La Réunion in the 9-10 March night, days 68 and 69 on Figures 17 and 18) when wet deposition would be expected to depress aerosol concentrations including PBAP though they recovered quickly following the storm. After the storm, PBAP concentrations vary episodically.

The fraction of total supermicron aerosol that was identified as fluorescent during daytime and nighttime periods is shown in Figure 16 along with average daytime and nighttime relative humidity. The fluorescent fraction was often higher at night than during the day and was not obviously related to variations in humidity. The highest fluorescent fractions were observed during the tropical storm Haliba (although the particle concentrations were very low) possibly indicating a small local source that persisted through the storm which would elevate the fluorescent fraction substantially due to the efficient removal of the vast majority of particles transported to the site. The low fluorescent fraction following Haliba is hypothesized to result from a sudden increase in non-biological anthropogenic aerosols following the storm when people on the island were finally able to drive after several days of significant road closures. After this initial post-storm depression, fluorescent fractions varied from 5-30%, similar to fractions that have been observed previously in continental locations.

The liquid water contents (LWC) of the collected clouds were low (mean value = 0.11 g m$^3$) and the period of cloud occurrence was most of the time less than 2 hours; for these reasons the collected liquid volumes were rather low (<15 ml) and did not allow a full chemical characterisation. However, the concentration of mono and di-carboxylic acids, formaldehyde, $H_2O_2$, iron and the main inorganic ions were quantified systematically. The mean dissolved organic carbon value was of 5.7 mgC.L$^{-1}$ which is close to value observed in mid-latitude sites such as the puy de Dôme station (5.5 mgC.L$^{-1}$ in average; Deguillaume et al., 2014). The main organic acid was the formic ones followed by acetic, oxalic and succinic. The inorganic ions were largely dominated by Cl$^-$ and Na$^+$, with a Cl$^-$/Na$^+$ ratio of 1.3, typical for area under marine influence. The seawater Cl/Na ratio is 1.5. Compared to this ratio, the marine aerosol has experienced some modification during emission to the atmosphere and activation to cloud droplets, with a substantial chloride depletion. This process was observed already on



primary marine aerosol (Schwier et al., 2017) indicating that further chloride depletion is likely not occurring after the aerosol has activated to cloud droplets.

For the microbial analysis, the energetic states of the cloud microflora given by the ADP/ATP ratio were not significantly higher than those evaluated for cloud samples collected at a mid-latitude site (puy de Dôme) (Vaïtilingom et al., 2012). The tropical location presents warmer temperature (mean=23°C) in comparison to the puy de Dôme (mean=4°C) but the microbial survival in clouds does not seem to be correlated with the temperature. The mean microbial cells content was of $3.8 \times 10^4$ cells mL$^{-1}$ which is lower but in the same order of magnitude than values from the puy de Dôme site ($5.4 \times 10^4$ cells mL$^{-1}$).

A total of 54 pure microbial strains have been isolated and identified (Figure 17); the main phylla found in La Réunion samples are *actinobacteria* (36%), *alpha* (20%) and *gamma* (17%)-*proteobacteria* and *firmicutes* (18%). Microbial strains that form these phylla are mainly associated with phyllosphere, soil and marine media. Main primary sources surrounding the Maïdo site therefore are from marine origin via the Indian Ocean and from biogenic origin through the dense forest cover. Bacteria from *firmicutes* and *actinobacteria* are twice more frequent in La Réunion samples than at the puy de Dôme (Vaïtilingom et al., 2012). This indicates an important role of microorganisms originated from the soil in the collected samples. It is also important to notice a high presence of phytopathogenic bacteria in these samples (full identification in progress).

## 5 Conclusions and perspectives

The FARCE campaign was designed to explore the forests-gases-aerosols-clouds system in the tropical island of La Réunion. It lasted from March the $6^{th}$ to April the $21^{st}$ 2015 and mainly focused on the area between the coast and the Maïdo Observatory. Figure 18 gives a compendious overview of the main processes occurring along the Maïdo mount slope that are related to the FARCE campaign. The goal was to improve our knowledge on the forest structure and $LAI$, concentrations of VOCs and precursors emitted by forests, aerosol loading and optical properties in the planetary boundary layer, formation of new particles by nucleation of gas-phase precursors, ice nucleating particles concentration, and biological loading in both cloud-free and cloudy conditions. This campaign required a significant human and scientific investment and a major effort for coordination of a multidisciplinary international team. These new results have been made possible by sustained efforts of cross-disciplinary collaborations between biologists, chemists and meteorologists, developing shared scientific language, methods and objectives during several years.

56 different plant species were identified within the forest plots with various species richness, together with highly different structures. $LAI^*$ values were estimated through four methods, which exhibit roughly similar estimates.

Isoprene concentrations were measured at all sites. Lowest concentrations of isoprene were found in the volcanic area with a very low vegetation density ($<25 \pm 9$ pptv), and the maxima of isoprene concentration were found within the forest plots ($200 \pm 68$ pptv in the Mare-Longue 150 m plot). Variation of the isoprene concentration was found to be well correlated to the variations of temperature and PAR. We observed a large diurnal variation of HCHO concentration at the Maïdo Observatory, which can be explained, during daytime, by the influence of boundary layer compounds associated to higher solar radiation and higher surface air temperatures, and, during nighttime, by the free tropospheric unperturbed marine environment.





Vertical distribution and optical properties of boundary layer aerosols were retrieved using a lidar in synergy with a handheld sun-photometer along the Maïdo mount slope. Measurements show that, between the shore and the Maïdo Observatory, the aerosol optical depth, the aerosol layer thickness and the aerosol average size decrease, while their absorption efficiency

increases, indicating that the sea-salt loading decreases as the lidar travels uphill toward the free troposphere.

The formation of new particles by nucleation of gas-phase precursors was observed almost every day of the FARCE campaign at the Maïdo Observatory. A typical measurement day shows that the aerosol concentration increases at the lowest size range from 03:00 UTC, progressively at higher diameters during the following hours and abruptly increases at around 06:00 UTC, showing that the boundary layer reaches the site. By late afternoon, the air mass is changing to a free tropospheric

particle-poor regime. The contribution of BVOCs to the formation and growth of new particles is an open question at this stage. The contribution of BVOCs to nucleation will be investigated by the comparison of the time evolution of nanoscale particle concentration with the time evolution of the BVOC concentrations. The contribution of BVOC to the growth of newly formed particles to larger sizes will be investigated by the comparison of the evolution of the condensational sink with the time evolution of the BVOC concentration (as performed in Sellegri et al., 2005). In order to exclude the contribution of any anthro-

pogenic species to the formation and growth of new particles, this will be explored using anthropogenic gas-phase tracers and black carbon.

INP concentrations and size were measured at the Maïdo Observatory, showing remarkably low average concentration values and higher average fraction of INPs > 1 $\mu$m compared to previous measurements in North America and Europe.

Supermicron PBAP loadings and fraction of total supermicron PBAP were measured at the Maïdo Observatory. Loading

measurements exhibit low to moderate concentrations in comparison to those reported for various continental locations, with higher concentrations during daytime than during nighttime. Both the fluorescent particle type distribution and observed number distributions are similar to what is observed for laboratory sampling of fungal spores, with substantial local biomass burning influence. Fluorescent aerosol concentrations were lowest and fluorescent fractions were highest when a strong tropical storm was close to La Réunion and, after the storm, fluorescent aerosol concentrations varied episodically, a finding which will be

further explored in a future work. More in-depth analysis of supermicron PBAP loadings and fraction of total supermicron PBAP data will be presented in a forthcoming manuscript. Local and regional bioaerosol sources will be examined using both observed meteorological parameters at the site and back trajectories. Both diurnal and multi-day episodic cycles in bioaerosol loadings observed at Maïdo will be investigated as well as correlations between observed bioaerosol and other parameters of interest such as formaldehyde, temperature and RH. To address questions regarding bioaerosol in the more regional remote

marine free troposphere, fluorescent particle concentrations will be assessed during periods of nighttime subsidence of free tropospheric air at the observatory identified manually based on observed abrupt changes in temperature, RH and ozone based on the supposition that FT-influenced air will be warmer, drier and have higher ozone than more locally influenced air. Fluorescent particle concentration data will be considered in conjunction with size-resolved observations of INP concentrations and genetic and component analysis of filter samples to identify key linkages between biological materials, observed fluorescent loadings and their potential impacts on cloud properties.





Clouds were collected at various locations during the FARCE campaign for chemical and microbial characterization. Mean dissolved organic carbon value is close to values observed in mid-latitude sites, and the inorganic ions concentrations are typical for area under marine influence. 54 pure microbial strains have been isolated and identified, and are mainly associated with phyllosphere, soil and marine media, confirming that main primary sources surrounding the Maïdo Observatory are from marine origin via the Indian Ocean and from biogenic origin through the dense forest cover. The description of the microbial content from these samples will be published in a forthcoming article and will be the first report on the bio-physico-chemistry of tropical clouds.

Preliminary numerical simulations confirm the main dynamics prevailing over the whole island and over the Maïdo mount slope. Simulated isoprene surface concentrations, wind direction, air temperature, and clouds agree well with observations on the Maïdo slope. Simulations show the existence of a horizontal windshear front located close to the Maïdo Observatory, which location drives the origin of the air masses that are sampled at the observatory and the vanishing of clouds. They also support the hypothesis of a potential way of SOA formation in aqueous phase at the Maïdo Observatory.

To sum up, these results corroborate the following conclusions:

- the Maïdo Observatory lies within the PBL from late morning to late evening;

- when in the PBL, the main primary sources impacting the Maïdo Observatory are from marine origin via the Indian Ocean and from biogenic origin through the dense forest cover.

New conclusions can be inferred from these results:

- the marine source prevails less and less while reaching the Observatory;

- when in the PBL, depending on the localization of a horizontal windshear front with respect to the Maïdo Observatory, this latter can be affected by air masses coming directly from the ocean and passing over the Maïdo mount slope, or coming from inland;

- bioaerosols can be observed in both dry air and clouds at the Maïdo Observatory;

- BVOCs emissions by the forest covering the Maïdo mount slope can be transported upslope within clouds and are a potential way of SOA formation in aqueous phase at the Maïdo Observatory;

- the simulation of dynamics parameters, emitted BVOCs and clouds life cycle in the Meso-NH model are realistic, and more advanced Meso-NH simulations should use an increased horizontal resolution (100 m) to better take into account the orography and improve the simulation of the windshear front zone within which lies the Maïdo Observatory.

These results largely contributed to the setting up of the OCTAVE and Biomaïdo projects. The ongoing OCTAVE project provides new material for the study of nucleation processes, as well as of sources and seasonal cycles of VOC and halogens from forests and Indian Ocean and for the quantification of the contribution of marine/biogenic sources to submicron organic aerosols at Maïdo. The starting Biomaïdo project focuses on the contribution of the aqueous reactivity to the SOA budget. The FARCE campaign provides a unique set of multi-disciplinary data and results that can be used as a reference socle for these projects to better understand the forest-gases-aerosols-clouds system in an insular tropical environment. These results will help to reduce uncertainties in the understanding and the modeling of the formation and transformation of atmospheric aerosols. This is needed to properly quantify the impacts of these particles on air quality, health and climate change.



*Data availability.*  TEXT

*Author contributions.*  All authors contributed significantly to this manuscript. Valentin Duflot wrote the manuscript draft with contributions from Pierre Tulet, Olivier Flores, Christelle Barthe, Aurélie Colomb, Laurent Deguillaume, Michael Vaïtilingom, Anne Perring, Alex Huff-

5  man and Karine Sellegri. All authors analyzed the data. Pierre Tulet, Olivier Flores, Christelle Barthe, Aurélie Colomb, Laurent Deguillaume, Michael Vaïtilingom, Anne Perring, Alex Huffman, Mark T. Hernandez, Ellis Robinson, Odessa Gomez, Frédéric Burnet, Thierry Bourri-anne, Jacques Fournel, Pierre Stamenoff, Jean-Marc Metzger, Mathilde Chabasset, Clothilde Rousseau, Eric Bourrianne and Valentin Duflot performed the experiments. All authors revised the manuscript draft and provided valuable suggestions for the revision.

*Competing interests.*  The authors declare that they have no conflict of interest.

10  *Acknowledgements.*  The authors acknowledge the European Communities, the Région Réunion, CNRS, and Université de la Réunion for their support and contributions in the construction phase of the research infrastructure OPAR (Observatoire de Physique de l'Atmosphère de La Réunion). OPAR is presently funded by CNRS (INSU) and Université de La Réunion and managed by OSU-R (Observatoire des Sciences de l'Univers de La Réunion, UMS 3365). The experiments have been funded by the Université de la Réunion through the federation Observatoire des Milieux Naturels et des Changements Globaux (OMNCG) of the OSU-R. The Service National d'Observation (SNO)

15  CLimate relevant Aerosol Properties from near surface observations (CLAP) and the ACTRIS-2 EU project are also acknowledged. This work was supported by the french national programme LEFE/INSU in the frame of the project entitled "Biophysicochimie des nuages tropicaux de l'Ile de la Réunion". Henri Legros receives our gratitude for the provision of an observation site.





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





**Figure 1.** Overview maps showing the location of La Réunion as well as of the FARCE campaign sites. Altitude and deployed instruments are given for each measurement site. The red line is the pick up track on the $2^{nd}$ of April 2015 (given as an example).



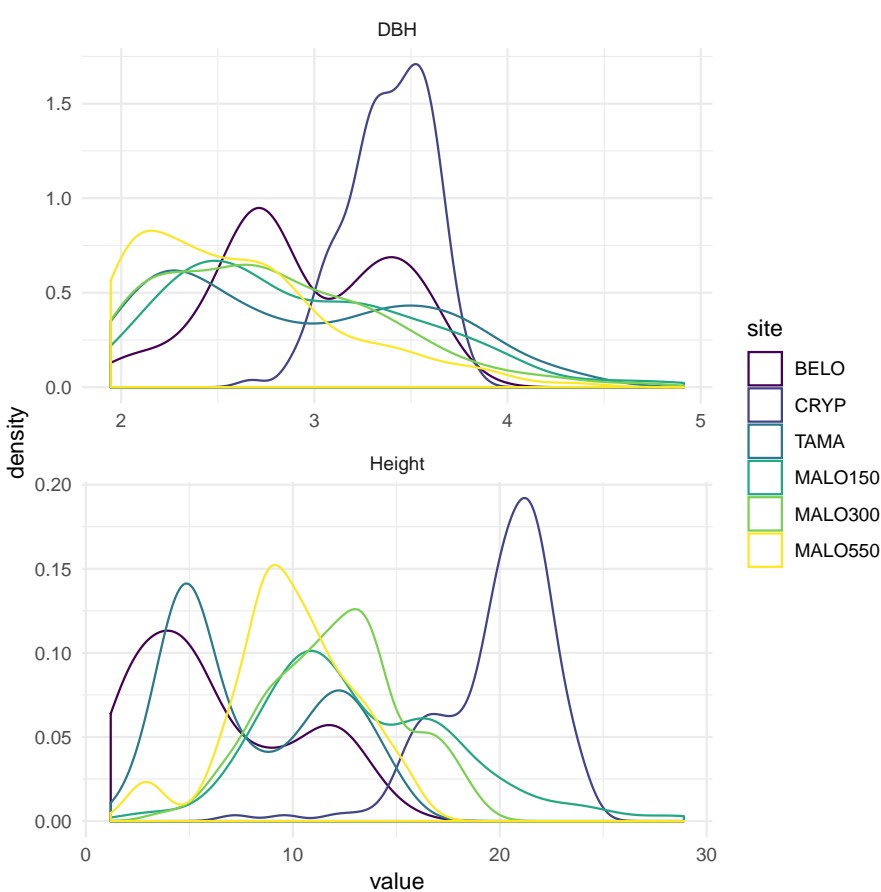

**Figure 2.** Density distribution of diameters (DBH, in dm, top panel) and heights (in m, bottom panel) in forest plots



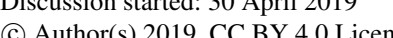


**Figure 3.** Allometric relationships between diameter and height in forest plots. Symbols indicate particular species or species set: ● *Acacia heterophylla* in BELO and TAMA, ∗ *Cyathea borbonica* and *Cyathea glauca* (tree-ferns) in BELO, ▲ *Cryptomeria japonica* in CRYP, ■ *Erica arborescens* in TAMA, ⊠ *Pandanus pupurescens* in MALO550, + other species. Smoothing lines are estimated without taking into account species with particular architecture (tree-ferns and *Pandanus pupurescens*).





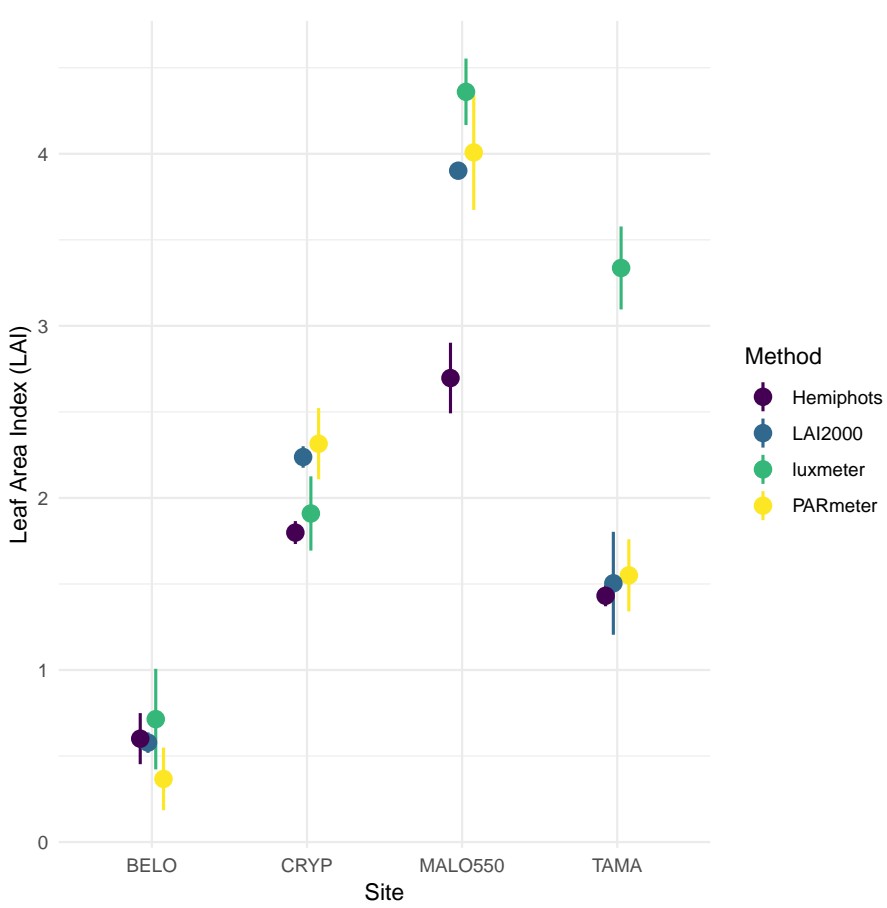

**Figure 4.** Estimates of Leaf Area Index (LAI) in four forest plots with four methods: hemiphots (hemispherical photographs), LAI2000, and light interception method, based on light intensity measurements using a luxmeter or a PARmeter. Points and vertical lines indicate average values and standard errors.





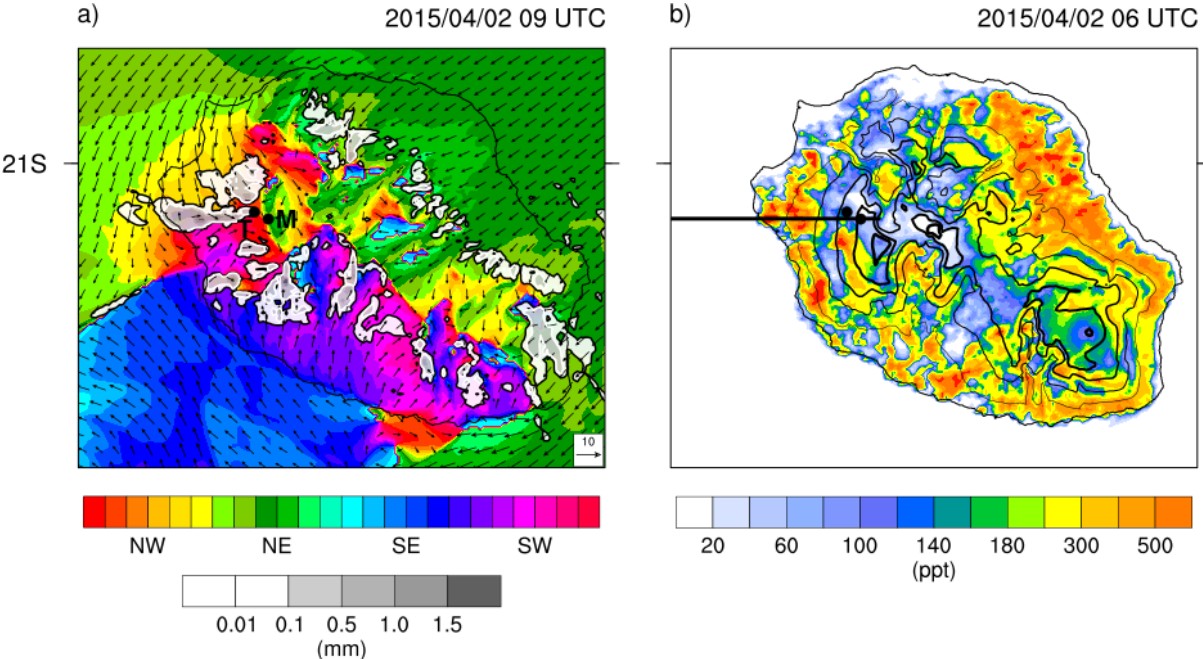

**Figure 5.** Meso-NH simulation for the $4^{th}$ of April 2015. Left panel: surface wind direction (color scale) and intensity (arrows in m.s$^{-1}$, reference given in the lower-right corner) and integrated cloud water content (gray shaded in mm) at 9:00 UTC. The letters T and M show the location of the Tamarins plot and Maïdo Observatory, respectively ; right panel: isoprene concentration at 6:00 UTC, black isolines show the orography, and black line shows the vertical cross-section shown on Figure 7.





**Figure 6.** Time series (UT) of the 10-m wind direction, 2-m temperature (°C), integrated cloud water content (mm) and isoprene concentration (pptv) at the Tamarins plot (simulated values in red dots) and at the Maïdo Observatory (simulated values in blue dots and observed values in crosses). Vertical bars show the dispersion of the simulated values on the model grid points surrounding both sites.





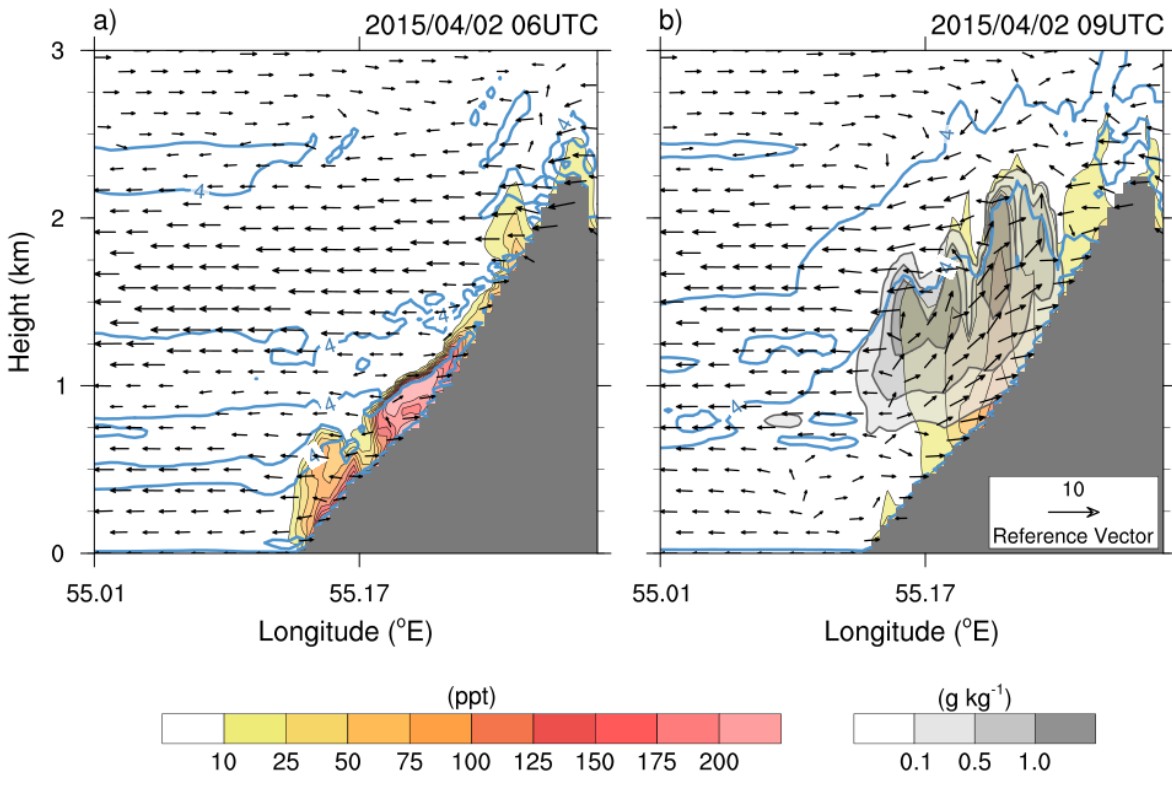

**Figure 7.** Vertical cross-section between 0 and 3 km asl along the line shown on Figure 5 (right panel) showing the isoprene concentration (color scale), cloud water (gray shaded in g.kg$^{-1}$) as well as the direction and intensity of the wind (in m.s$^{-1}$, scale given in the lower right corner) simulated by the Meso-NH model for the $2^{nd}$ of April 2015 at 6:00 UTC (left panel) and 9:00 UTC (right panel). Blue isolines show the square equivalent Brunt-Väisälä frequency (isoline at $4 \times 10^{-4}$s$^{-2}$).





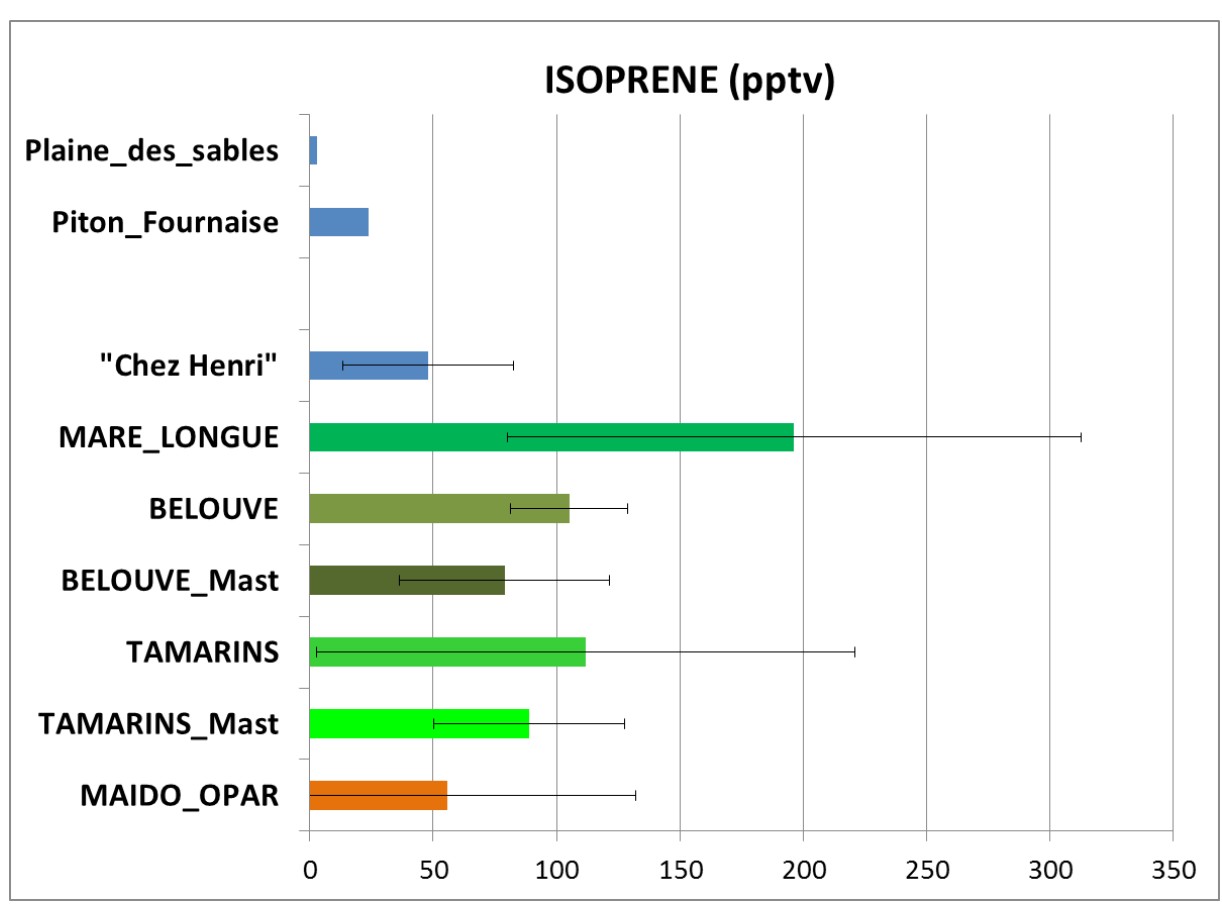

**Figure 8.** Mean and standard deviation of Isoprene mixing ratio (in pptv) for all sampling sites during the campaign.



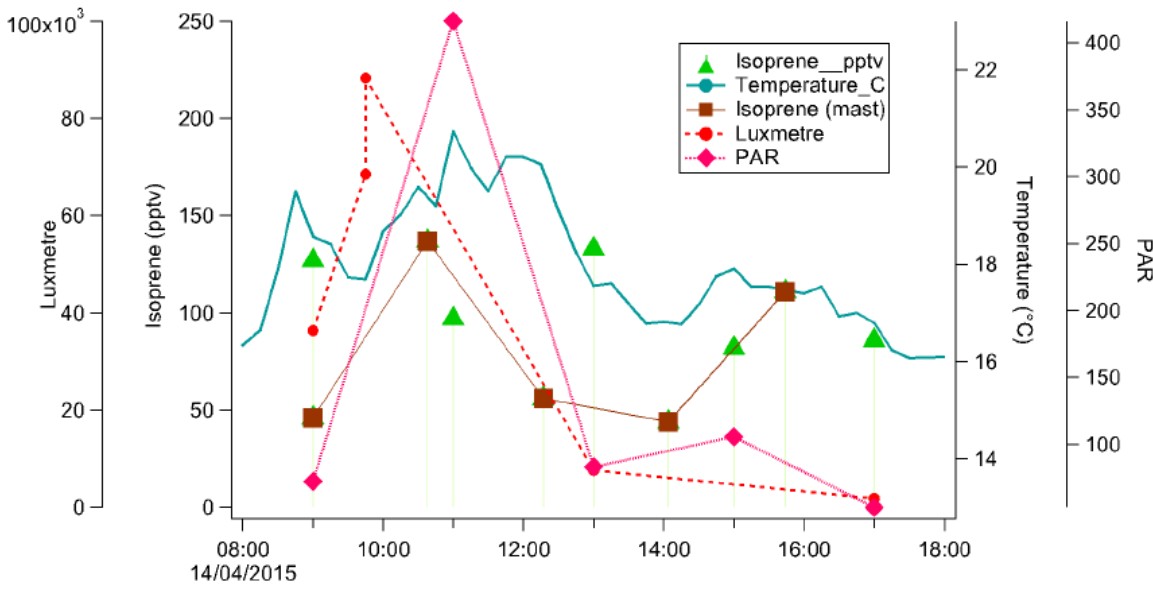

**Figure 9.** Diurnal variation of isoprene (pptv) with PAR ($\mu$mol.s$^{-1}$.m$^{-2}$) and temperature ($^\circ$C) in the Bélouve forest, on the $14^{th}$ of April 2015. X axis gives Local Time.

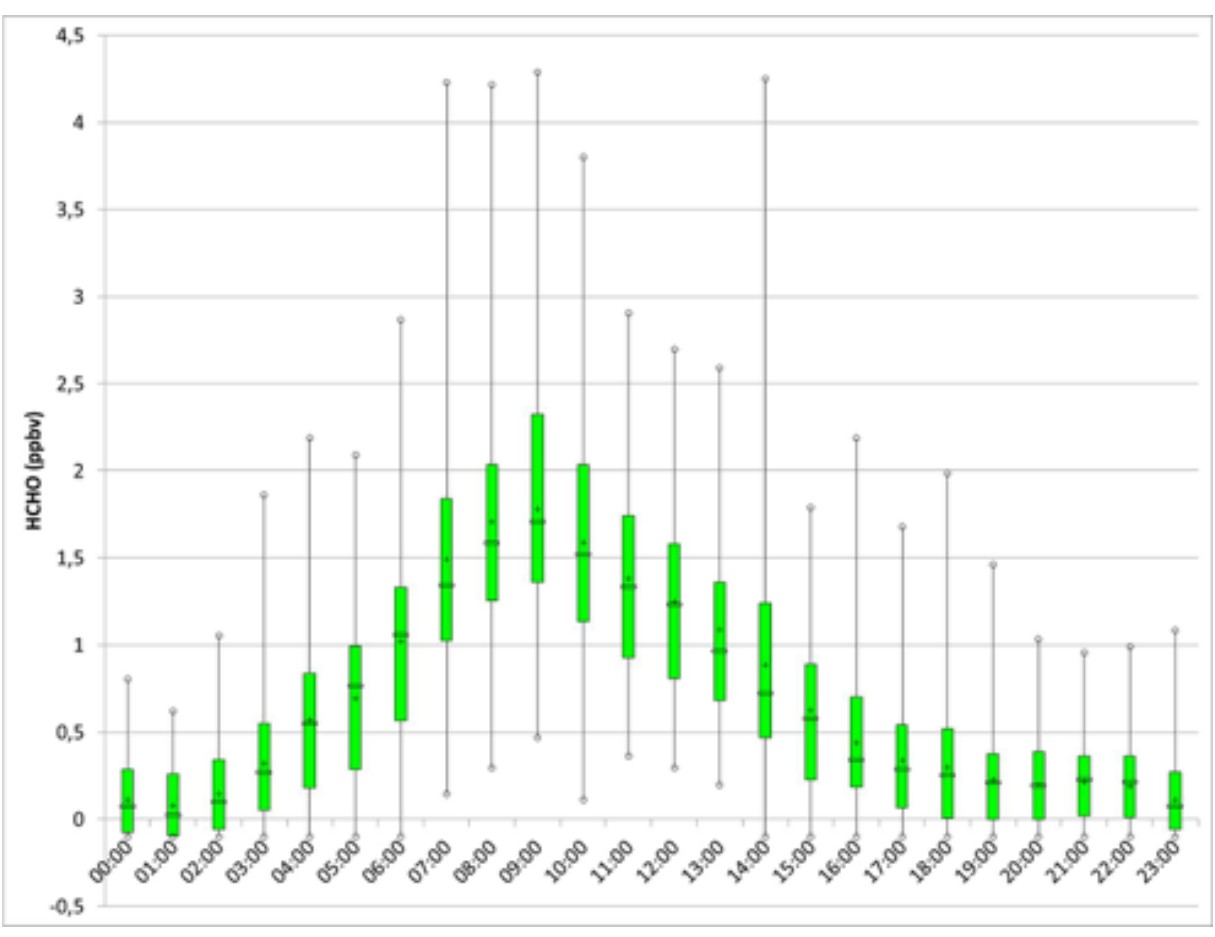

**Figure 10.** Mean diurnal Variation of Formaldehyde (ppbv) at the Maïdo Observatory. Measurements were performed from the $11^{th}$ of March to the $11^{th}$ of April 2015. Time is UTC.





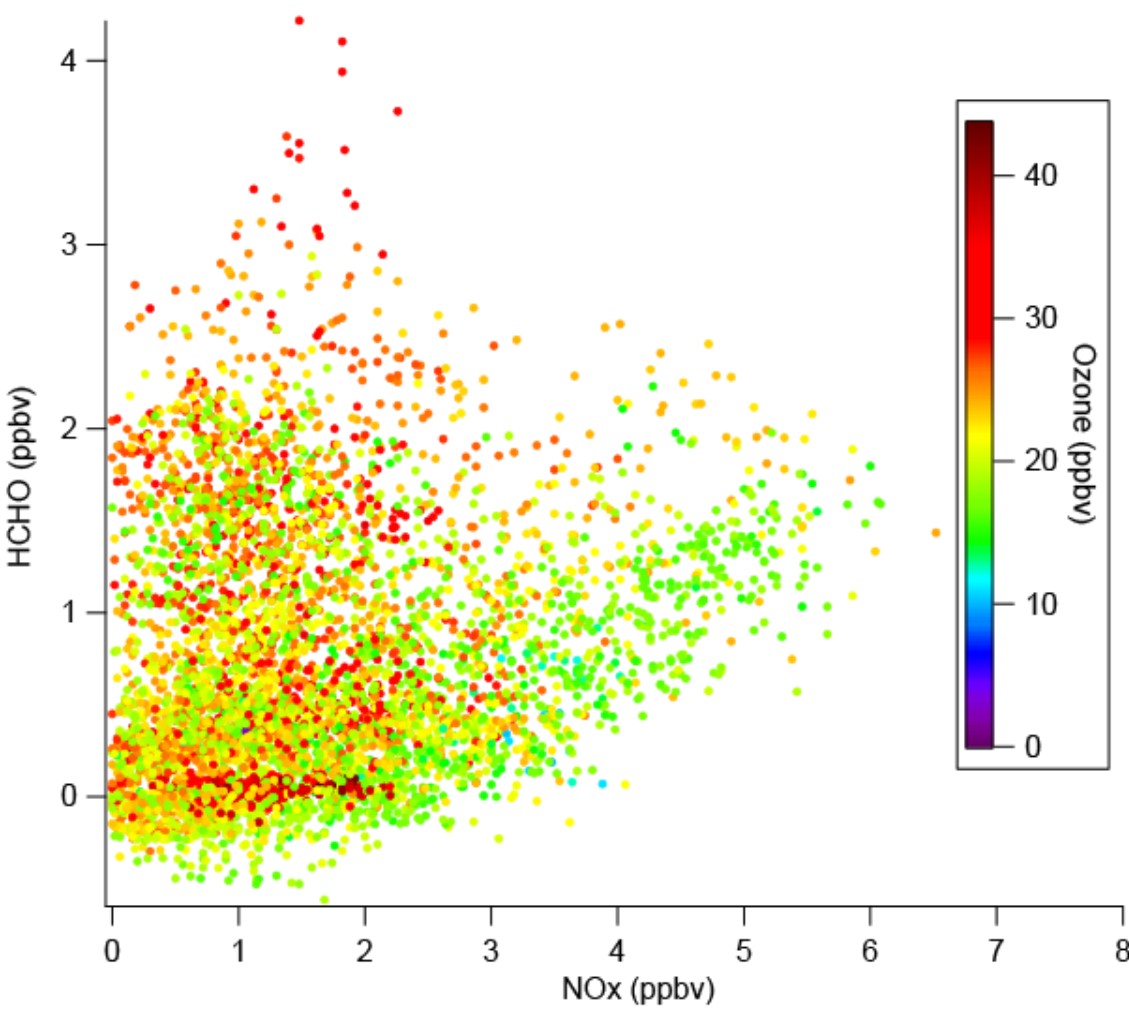

**Figure 11.** HCHO versus NOx (ppbv) and colored with ozone concentration (ppbv). Measurements were performed from the 11[th] of March to the 11[th] of April 2015 at the Maïdo Observatory.





**Figure 12.** (top panel) Map showing the track of the pick up carrying the mobile aerosol lidar from Saint Paul up to the Maido station on the $2^{nd}$ of April 2015. Campaign sites are shown similarly to Figure 1. (bottom panel) Lidar backscattered signals along the lidar track. Colored crosses (top panel) and dotted lines (bottom panel) show the location of the lidar corresponding to the extinction profiles shown on Figure 13.



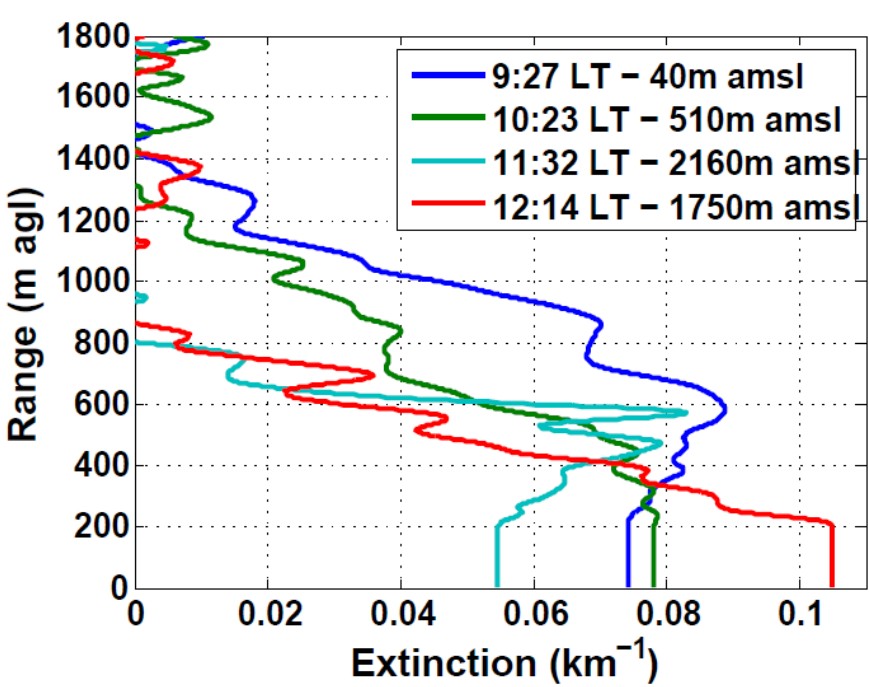

**Figure 13.** Extinction profiles corresponding to the lidar observations shown on Figure 12.





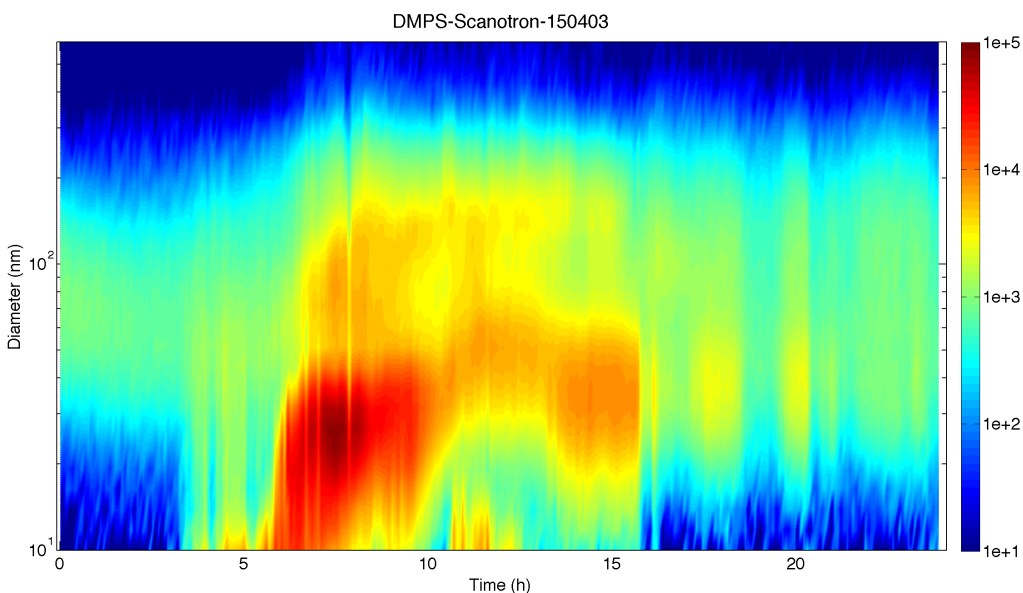

**Figure 14.** Daily variation of the aerosol size distribution (10-550 nm) observed on the $3^{rd}$ of April 2015. The aerosol concentration (dN/dLogDp) is illustrated with the colour bar, time is on the x axis (UTC) and aerosol diameter is on the y axis.





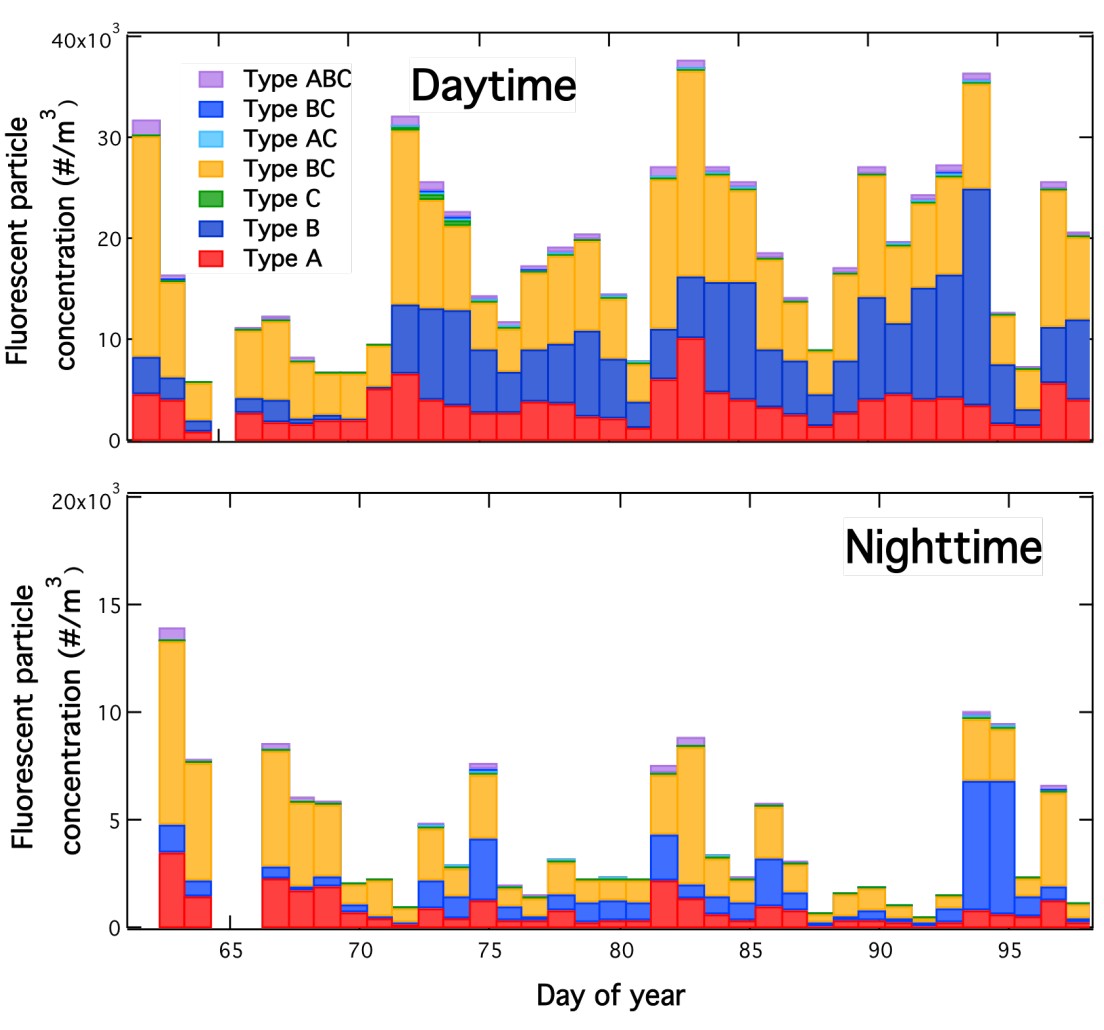

**Figure 15.** Daytime (top panel) and nighttime (bottom panel) fluorescent particle concentrations observed at the Maïdo Observatory.





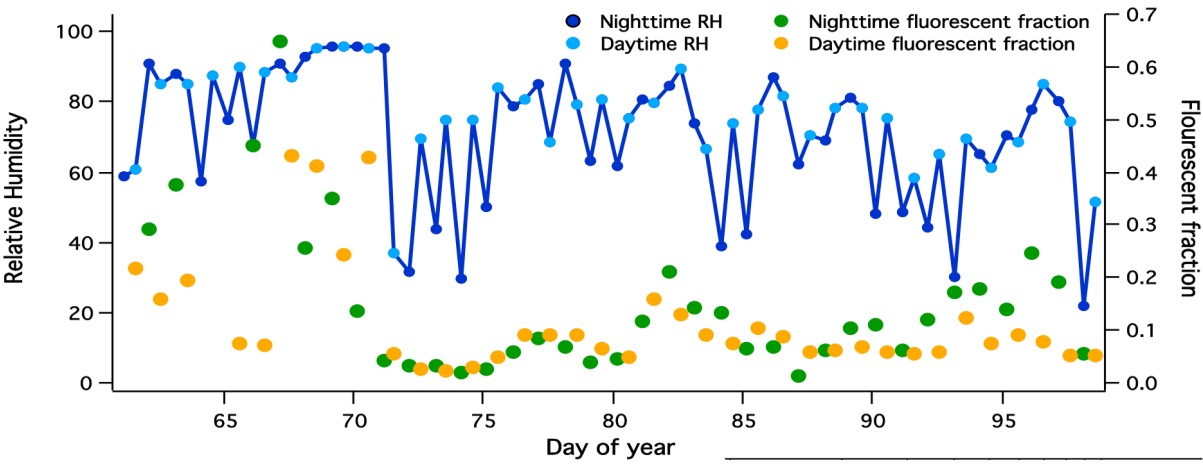

**Figure 16.** Relative humidity (dark blue dots for nighttime and light blue dots for daytime) and supermicron fluorescent fraction observed throughout the study period (green dots for nighttime and orange dots for daytime) at the Maïdo Observatory.



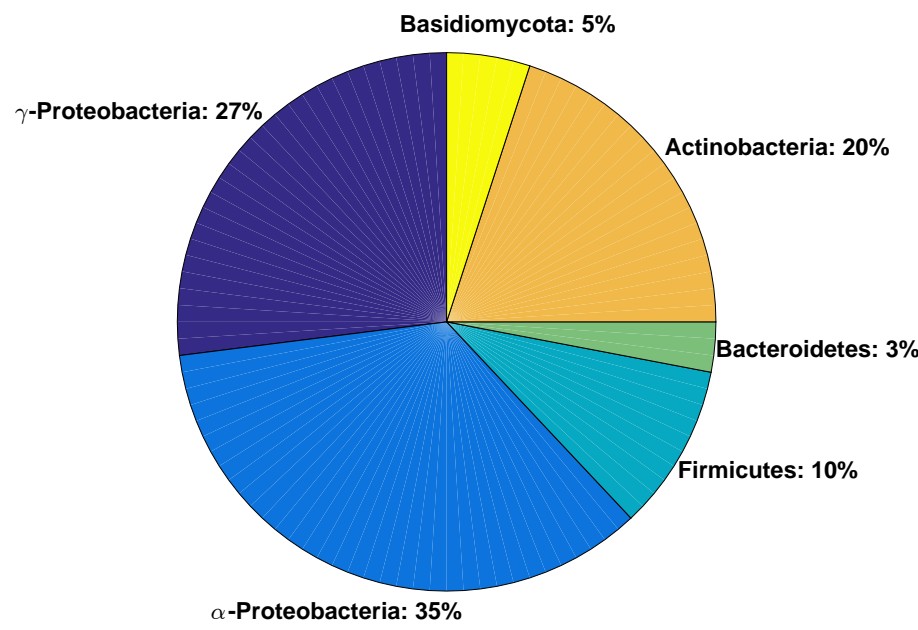

**Figure 17.** Relative abundance of microbial phylla in cloud water from La Réunion.



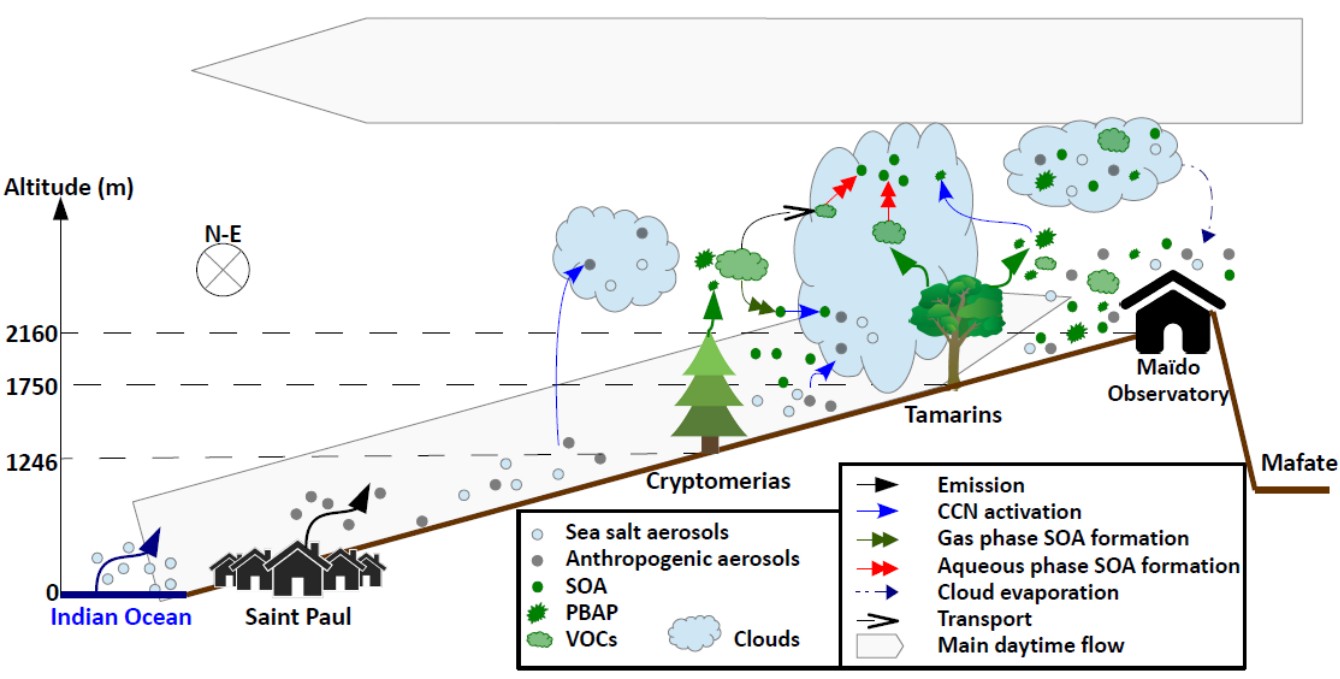

**Figure 18.** Compendious overview of the main processes occurring along the Maïdo mount slope that are related to the Reunion FARCE campaign.

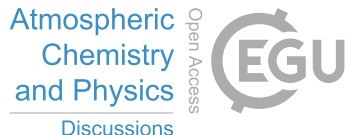

| Site/Measurements | Cryptomerias | Chez Henri | Tamarins | Maïdo station | Pick up | Bélouve | Mare Longue | Plaine des sables | Piton de la Fournaise |
|---|---|---|---|---|---|---|---|---|---|
| Latitude; Longitude; Altitude | 21.08°S; 55.34°E; 1248m | 21.05°S; 55.35°E; 1305m | 21.07°S; 55.36°E; 1750m | 21.08°S; 55.38°E; 2160m | - | 21.06°S; 55.54°E; 1520m | 21.35°S; 55.74°E; 150, 330, 550m | 21.23°S; 55.65°E; 2265m | 21.22°S; 55.68°E; 2360m |
| VOC | 7/4, 8/4*, 9/4, 10/4 | 26/3+, 27/3+, 31/3+, 1/4+, 9/4+, 10/4+ | 2/4*, 7/4, 9/4, 10/4 | 12/3-10/4 | | 14/4* | 21/4 (150m) | 6/4 | 29/3 |
| PAR | 7/4, 8/4*, 9/4, 10/4 | | 2/4*, 7/4, 9/4, 10/4 | | | 14/4* | 21/4 (150m) | | |
| LAI | 8/4 | | 2/4 | | | 14/4 | 21/4 (550m) | | |
| Species, TTH and DBH | 8/4 | | 2/4 | | | 14/4 | 20-30/4 | | |
| Cloud impactor | | 27/3, 8/4, 9/4 | 26/3 | | | | | | |
| Lidar | | | | | 1/4, 2/4, 9/4 | | | | |
| Sun photometer | | | | | 1/4, 2/4, 9/4 | | | | |
| DMPS | | | | 6/3-10/4 | | | | | |
| WIBS | | | | 6/3-10/4 | | | | | |
| MOUDI | | | | 6/3-10/4 | | | | | |
| NOx analyser | | | | 6/3-10/4 | | | | | |
| O3 analyser | | | | 6/3-10/4 | | | | | |
| HCHO analyser | | | | 6/3-10/4 | | | | | |
| Meteo Station | 7/4, 8/4, 9/4, 10/4 | 26/3, 27/3, 31/3, 1/4, 8/4, 9/4, 10/4 | 26/3, 2/4, 7/4, 9/4, 10/4 | 6/3-10/4 | | 14/4 | 21/4 | | |

**Table 1.** Summary and dates (day/month) of the measurements performed at each site. *: Measurements performed along the day. +: Measurements performed on top of the 10 m mast.



| Site | n | S | BA | DBH.mean | DBH.sd | DBH.max | H.mean | H.sd | H.max |
|------|---|---|-----|----------|--------|---------|--------|------|-------|
| BELO | 123 | 5 | 4.9 | 21 | 9.1 | 45 | 6.4 | 3.9 | 15 |
| CRYP | 151 | 2 | 11.2 | 30 | 6.2 | 46 | 19.9 | 2.7 | 24 |
| TAMA | 149 | 2 | 9.2 | 23 | 16.2 | 79 | 7.9 | 3.8 | 15 |
| MALO150 | 165 | 25 | 11.1 | 23 | 18.6 | 136 | 13.3 | 4.6 | 29 |
| MALO300 | 250 | 26 | 11.7 | 20 | 14.8 | 111 | 12.1 | 3.2 | 19 |
| MALO550 | 363 | 34 | 10.5 | 16 | 10.9 | 83 | 10.0 | 3.0 | 17 |

**Table 2.** Summary statistics for tree stands within sampled forest plots. n: number of measured stems, S: species richness, BA: basal area ($m^2.ha^{-1}$) plus mean, standard deviation (sd) and maximal values for tree DBH (cm) and height (m).



| LT (UTC) | $Z_l$ (m asl) | $D_s$ (km) | $AOT_{355}$ | $\mathring{A}$ | $LR_{355}$ (sr) | $H_a$ (m agl) |
|---|---|---|---|---|---|---|
| 9:27 (13:27) | 40 | ≈1 | $0.08 \pm 0.02$ | $1.06 \pm 0.16$ | $30 \pm 8$ | ≈1400 |
| 10:23 (14:23) | 510 | ≈4 | $0.07 \pm 0.02$ | $1.11 \pm 0.17$ | $41 \pm 10$ | ≈1300 |
| 11:32 (15:32) | 2160 (Maïdo Obs.) | ≈14 | $0.03 \pm 0.01$ | $1.38 \pm 0.21$ | $50 \pm 13$ | ≈800 |
| 12:14 (16:14) | 1750 (Tamarins plot) | ≈12 | $0.05 \pm 0.01$ | $1.30 \pm 0.20$ | $88 \pm 22$ | ≈850 |

**Table 3.** Lidar altitude ($Z_l$) and distance from the shore ($D_s$), $AOT_{355}$, Ångström coefficient between 500 and 870nm ($\mathring{A}$), retrieved LR at 355nm ($LR_{355}$), and vertical extension of the aerosol layer ($H_a$) for the 4 extinction profiles shown on Figures 15 and 16.





| Freezing temp. (°C) | Mean INP conc. (L$^{-1}$) | | Fraction INPs>1.8$\mu$m | Fraction INPs>1.0$\mu$m | | No. samples INP>0 |
|---|---|---|---|---|---|---|
| | Réunion | Mason et al., 2016 | Réunion | Réunion | Mason et al., 2016 | |
| -30 | $3.0 \pm 2.6$ | nr | $0.70 \pm 0.18$ | $0.79 \pm 0.20$ | nr | 23/23 |
| -25 | $0.20 \pm 0.21$ | 1-10 | $0.86 \pm 0.33$ | $0.89 \pm 0.32$ | 0.40-0.70 | 18/23 |
| -20 | $0.022 \pm 0.068$ | 0.2-2 | $1.0 \pm 0.0$ | $1.0 \pm 0.0$ | 0.52-1.0 | 3/23 |

**Table 4.** Overview of INP measurements from FARCE campaign. INP concentration and INP fraction at two particle size cut-point shown as mean values $\pm$ standard deviation. The range of observed values from six measurement locations summarized by Mason et al. (2016) are shown for comparison. Values at -30°C are not reported (nr) by Mason et al. (2016). Number of samples showing measurable INP shown.