# Peer review of "Preliminary results from the FARCE 2015 campaign: multidisciplinary study of the forests-gases-aerosols-clouds system on the tropical island of La Réunion"

_Atmospheric Chemistry and Physics, 2019_

## Referee Comment (RC1) · Anonymous Referee #1 · 28 May 2019

The authors state that this manuscript "intends to describe the Reunion FARCE campaign set up, and to provide the preliminary results...". While not a traditional scientific paper, it is understandable that this kind of papers are written in relation to large research projects. My problem here is that FARCE appears not to be a large research project, but rather an exploratory field campaign somehow related to two other, larger projects that are either ongoing (OCTAVE) or to be started soon (Biomaido). It remains unclear to me whether the preliminary results reported here are relevant only the FARCE campaign, or whether this paper could also serve as a basis for the two

other projects. In the former case, I am a bit skeptical about the usefulness of this paper. In any case, I think that this paper should be shortened to some extent, as suggested below. An addition to this, I have a few scientific and technical issues that should be taken care of before accepting this paper for publication.

Suggestions for shortening the paper

The introduction of this paper is relatively well written, but it also contains review-type material (e.g. SOA formation pathways) that has very little to do with the current paper. I would recommend removing some of this material from the paper.

Section 3 is well written but rather long. If this section is only for the purpose of this paper, it should be shortened. If it is meant to be a reference for later papers related to the 2 large research projects conducted at this site, then this section is acceptable as it is.

Concerning the preliminary results, I do not see section 4.4 useful at all. Sections 4.3 and 4.5.3 seem overly long to me.

I do not see Figure 10 or 11 useful.

Scientific issues

Page 2, line 28: What is meant by "primary cloud particles"?

Page 15, lines 11-15 and page 16, lines 17-18. Based on Figures 6 and 9, it is not correct to claim that measured and modeled isoprene concentrations are in good (or remarkable) agreement each other. Also, nothing can be said about the correlation between different quantities based on so few measurement points for isoprene.

Technical issues

Pge 18, line 25: For contrast -> In contrast

Several of the figures are either technically poor (e.g. missing axises in figs. 2 and 3)

or have a figure caption short of information about the contents of the figure (in figure 6, there is even something wrong in the description of the two lowest figure panels).

---

## Referee Comment (RC2) · Anonymous Referee #2 · 25 Jun 2019

The article by Duflot et al. describes some results from a field campaign conducted on the island of La Réunion. It provides a set of useful information to better understand the interactions in the forest-gases-aerosols-clouds system, and such measurements are needed and potentially important. However, these results are preliminary in this manuscript and will be further discussed in future. The main concern here is that the FARCE campaign is only an exploratory campaign to the other two ongoing/pending campaigns, which makes the importance of publishing the current "preliminary" results questionable as a scientific paper.

[Figure]

There are also other issues need to be addressed, as listed below:

Page 11, lines 31: Why UTC is used in this section and all the figures, but local time is used in Section 4.4 and after?

Page 15, line 10: The comparison between observation and model simulation does not show a "Remarkably well" agreement in Figure 6.

Page 16, lines 28-31: Case studies are needed to have a solid conclusion here, Figure 11 just contains too many data points. For instance, there are also high HCHO and significant ozone concentrations in the high NOx region shown in this figure.

Figure 5: Axis labels are missing.

---

## Author Comment (AC1) · 19 Jul 2019

Dear Editor,

We first wish to thank the reviewers for their constructive comments, which helped us to improve the article. Please find below answers to reviewer #1's comments and changes made in the manuscript "Preliminary results from the FARCE 2015 campaign: multidisciplinary study of the forests–gases–aerosols–clouds system on the tropical island of La Réunion" by V. Duflot et al.
- The authors state that this manuscript "intends to describe the Reunion FARCE campaign set up, and to provide the preliminary results...". While not a traditional scientific paper, it is understandable that this kind of papers are written in relation to large research projects. My problem here is that FARCE appears not to be a large research project, but rather an exploratory field campaign somehow related to two other, larger projects that are either ongoing (OCTAVE) or to be started soon (Biomaido). It remains unclear to me whether the preliminary results reported here are relevant only the FARCE campaign, or whether this paper could also serve as a basis for the two other projects. In the former case, I am a bit skeptical about the usefulness of this paper.

It is true that this article is not a "traditional scientific paper", and we especially thank the reviewer for addressing this point and helping us to clarify the "scientific positioning" of this work. This paper aims to complete the work of Baray et al. (AMT 2013), which gives an overview of the scientific potentiality of the Maïdo facility but focusing especially on remote sensing instruments and free tropospheric, UTLS and stratospheric matters. Our work intends to promote the Maïdo observatory's scientific specificities and potentialities for studies dealing with PBL processes occurring in a tropical insular environment. It draws up an inventory of the in situ studies that could be performed in this recent atmospheric observatory using various observations and simulations to better characterize the site. It has also vocation to develop scientific collaborations and to support future scientific programs, such as OCTAVE and Biomaïdo, whose related papers will use the results presented in this work to build up their discussions and conclusions. It is now clarifed in the text (Abstract, p. 2, l. 10-13; Introduction, p. 5, l. 11-19; Conclusion, p.22 l. 33-35 and p.23 , l. 1-11).

- Suggestions for shortening the paper The introduction of this paper is relatively well written, but it also contains review-type material (e.g. SOA formation pathways) that has very little to do with the current paper. I would recommend removing some of this material from the paper.

We believe it is valuable for the reader to expose the general context, the specific concerns and questions remaining open that prompted the organization of the FARCE project (and, consequently, the OCTAVE and Biomaïdo ones), as this project gives some pieces of the answers to these open questions. This introduction aims at providing the state-of-the-art for each of the aspects of the forests–gases–aerosols–clouds system (which are numerous, which explains partly the length of the introduction) and intends to be as exhaustive as possible in order to be used in the upcoming papers (dealing with FARCE, OCTAVE and Biomaïdo projects). This is the reason why it contains some review-type material and may look like a small ÂńÂăWhite PaperÂăÂż.

- Section 3 is well written but rather long. If this section is only for the purpose of this paper, it should be shortened. If it is meant to be a reference for later papers related to the 2 large research projects conducted at this site, then this section is acceptable as it is.

Section 3 describes the methods, measurements and model used in this study. Its length is partly explained by the number and diversity of the instruments involved. Moreover, Section 3 should indeed be used by the upcoming papers for the instruments used in common.

- Concerning the preliminary results, I do not see section 4.4 useful at all.

It is essential to document what (for both gases and particles) comes from the marine environment and is sampled at the measurement sites after traveling along the Maïdo mount slope. Section 4.4 gives the optical properties of the boundary layer aerosols and allows a validation of the Meso-NH simulations. Information on the optical properties of the boundary layer aerosols gives an insight on the type of aerosols (sea salt vs combustion-induced particles) reaching the measurement sites. The detection of clouds by lidar is used to evaluate the capacity of Meso-NH to simulate the cloud presence on a specific case (Section 4.3, p.16, l.1-3).

- Sections 4.3 and 4.5.3 seem overly long to me.

Sorry about that. However, we do not see how we could shorten these sections.

- I do not see Figure 10 or 11 useful.

As stated Section 3.2 (p.7-8, l.28-3), (B)VOCs, NOx, O3 and HCHO are deeply linked within the boundary layer, especially in tropical forests. Figure 10 gives the diurnal variation of HCHO concentration at the Maïdo Observatory, which provides information on the VOCs daily cycle (secondary production of HCHO is initiated in the continental boundary layer by the oxidation of VOCs) (Section 4.3, p.16, l.18-24). Figure 11 is the scatterplot of HCHO vs NOx vs O3 concentrations at the Maïdo Observatory. It helps identifying 2 situations driving the concentration of HCHO: VOCs photooxidation vs anthropogenic emissions (HCHO is directly emitted in the atmosphere by biomass burning, traffic and industrial emissions) (Section 4.3, p.16, l.25-33).

- Scientific issues Page 2, line 28: What is meant by "primary cloud particles"?

The word "primary" is indeed not relevant ; it has been removed from the text.

- Page 15, lines 11-15. Based on Figures 6 and 9, it is not correct to claim that measured and modeled isoprene concentrations are in good (or remarkable) agreement each other.

Section 4.3 states: "At both sites, the range of simulated isoprene concentrations agree remarkably well with the observations. [. . .] Taking into account error bars and standard deviations, one can see that there is an overall agreement between the measured and simulated time series of isoprene concentrations at both sites." We agree this is a bit confusing as we intend to differentiate the comparison of concentrations between observations and simulations for, on one hand, the range (which agrees remarkably well) and, on the other hand, the times series (which exhibits an overall agreement). We therefore removed the sentence: "At both sites, the range of simulated isoprene concentrations agree remarkably well with the observations".

- Page 16, lines 17-18. Nothing can be said about the correlation between different

quantities based on so few measurement points for isoprene.

This is right: correlation is not the appropriate word for so few measurement points. We changed the sentence into: "One can see that the variations of these three parameters follow the same pattern." (p.16, l.16-17).

- Technical issues Pge 18, line 25: For contrast -> In contrast

Corrected.

- Several of the figures are either technically poor (e.g. missing axises in figs. 2 and 3) or have a figure caption short of information about the contents of the figure (in figure 6, there is even something wrong in the description of the two lowest figure panels).

This is right for Figures 2 and 6 (axises are on Figure 3). Axises in Figures 2 have been added and caption of Figure 6 has been corrected. Moreover, Figure 10 has been remade with a better resolution.

Please also note the supplement to this comment:
https://www.atmos-chem-phys-discuss.net/acp-2019-341/acp-2019-341-AC1-supplement.pdf

---

## Author Comment (AC2) · 19 Jul 2019

Dear Editor,

We first wish to thank the reviewers for their constructive comments, which helped us to improve the article. Please find below answers to reviewer #2's comments and changes made in the manuscript "Preliminary results from the FARCE 2015 campaign: multidisciplinary study of the forests–gases–aerosols–clouds system on the tropical island of La Réunion" by V. Duflot et al.

[Figure]

- The article by Duflot et al. describes some results from a field campaign conducted on the island of La Réunion. It provides a set of useful information to better understand the interactions in the forest-gases-aerosols-clouds system, and such measurements are needed and potentially important. However, these results are preliminary in this manuscript and will be further discussed in future. The main concern here is that the FARCE campaign is only an exploratory campaign to the other two ongoing/pending campaigns, which makes the importance of publishing the current "preliminary" results questionable as a scientific paper.

It is true that this article is not a "traditional scientific paper", and we especially thank the reviewer for addressing this point and helping us to clarify the "scientific positioning" of this work. This paper aims to complete the work of Baray et al. (AMT 2013), which gives an overview of the scientific potentiality of the Maïdo facility but focusing especially on remote sensing instruments and free tropospheric, UTLS and stratospheric matters. Our work intends to promote the Maïdo observatory's scientific specificities and potentialities for studies dealing with PBL processes occurring in a tropical insular environment. It draws up an inventory of the in situ studies that could be performed in this recent atmospheric observatory using various observations and simulations to better characterize the site. It has also vocation to develop scientific collaborations and to support future scientific programs, such as OCTAVE and Biomaïdo, whose related papers will use the results presented in this work to build up their discussions and conclusions. It is now clarifed in the text (Abstract, p. 2, l. 10-13; Introduction, p. 5, l. 11-19; Conclusion, p.22 l. 33-35 and p.23 , l. 1-11).

- There are also other issues need to be addressed, as listed below: Page 11, lines 31: Why UTC is used in this section and all the figures, but local time is used in Section 4.4 and after?

Good point. There is no reason for that. Local times have been changed into UTC times.

- Page 15, line 10: The comparison between observation and model simulation does not show a "Remarkably well" agreement in Figure 6.

Section 4.3 states: "At both sites, the range of simulated isoprene concentrations agree remarkably well with the observations. [. . .] Taking into account error bars and standard deviations, one can see that there is an overall agreement between the measured and simulated time series of isoprene concentrations at both sites." We agree this is a bit confusing as we intend to differentiate the comparison of concentrations between observations and simulations for, on one hand, the range (which agrees remarkably well) and, on the other hand, the times series (which exhibits an overall agreement). We therefore removed the sentence: "At both sites, the range of simulated isoprene concentrations agree remarkably well with the observations".

- Page 16, lines 28-31: Case studies are needed to have a solid conclusion here, Figure 11 just contains too many data points. For instance, there are also high HCHO and significant ozone concentrations in the high NOx region shown in this figure.

That's right. Figure 11 only gives an first insight in the determination of the sources of HCHO at the Maïdo Observatory. A paper dedicated to this topic is in preparation. Figure 11 has been modified to better show the 2 kinds of situation: high O3/low NOx, and low ozone/high NOx, and the text better states the fact that these conclusions are preliminary and need further studies (p. 16, l. 25-33).

- Figure 5: Axis labels are missing.

Right. Axis labels were added on Figure 5.

Please also note the supplement to this comment:
https://www.atmos-chem-phys-discuss.net/acp-2019-341/acp-2019-341-AC2-supplement.pdf
* * *
[Figure]

2019.

**Supplement:**

[revised manuscript text omitted]

---

## Author Comment (AC3) · 19 Jul 2019

Dear Editor,

I would like to add a coauthor to this article who I forgot to include in the initial submission phase: Manon Rocco. She indeed contributed to this study by analyzing BVOCs, HCHO, NOx and ozone measurements, and by revising the manuscript.

Here is her email address: rocco.manon@gmail.com.

[Figure]

I would be grateful if you could include her as a coauthor fo this article (her name and affiliation is in the new version of the manuscript, which I attach here as a supplement).

Thank you and best regards, Valentin.

Please also note the supplement to this comment:
https://www.atmos-chem-phys-discuss.net/acp-2019-341/acp-2019-341-AC3-supplement.pdf